# Site-directed M2 proton channel inhibitors enable synergistic combination therapy for rimantadine-resistant pandemic influenza

**Claire Scott**[1,2¤a], **Jayakanth Kankanala**[2,3¤b], **Toshana L. Foster**[1,2¤c], **Daniel H. Goldhill**[4], **Peng Bao**[5], **Katie Simmons**[2,3], **Marieke Pingen**[1¤d], **Matthew Bentham**[1,2], **Elizabeth Atkins**[1,2¤e], **Eleni Loundras**[1,2], **Ruth Elderfield**[4¤f], **Jolyon K. Claridge**[6¤g], **Joseph Thompson**[2,3¤h], **Peter R. Stilwell**[4¤i], **Ranjitha Tathineni**[1¤j], **Clive S. McKimmie**[1], **Paul Targett-Adams**[7¤k], **Jason R. Schnell**[6], **Graham P. Cook**[1], **Stephen Evans**[5], **Wendy S. Barclay**[4☯], **Richard Foster**[2,3☯], **Stephen Griffin**[1,2]*

**1** Leeds Institute of Medical Research, School of Medicine, Faculty of Medicine and Health, St James's University Hospital, University of Leeds, Leeds, West Yorkshire, United Kingdom, **2** Astbury Centre for Structural Molecular Biology, University of Leeds, Leeds, West Yorkshire, United Kingdom, **3** School of Chemistry, Faculty of Mathematics and Physical Sciences, University of Leeds, Leeds, West Yorkshire, United Kingdom, **4** Faculty of Medicine, Section of Virology, Imperial College, London, United Kingdom, **5** Faculty of Engineering and Physical Sciences, School of Physics and Astronomy, University of Leeds, Leeds, West Yorkshire, United Kingdom, **6** Department of Biochemistry, University of Oxford, Oxford, United Kingdom, **7** Pfizer UK, Sandwich, Kent, United Kingdom

☯ These authors contributed equally to this work.
¤a Current address: Covance, Springfield House, West Yorkshire, United Kingdom
¤b Current address: Centre for Drug Design, University of Minnesota, Nils Hasselmo Hall, Minneapolis, United States of America
¤c Current address: Faculty of Medicine & Health Sciences, University of Nottingham, Sutton Bonington Campus, Sutton Bonington, Leicestershire, United Kingdom
¤d Current address: Institute of Infection, Immunity and Inflammation, College of Medical, Veterinary and Life Sciences, University of Glasgow, Glasgow, United Kingdom
¤e Current address: London School of Hygiene & Tropical Medicine, London, United Kingdom
¤f Current address: Public Health England, Porton Down, Wiltshire,
¤g Current address: Structural and Molecular Microbiology, Structural Biology Research Centre, Vrije Universiteit Brussel, Pleinlaan, Brussels, Belgium
¤h Current address: School of Medicine, King's College London, Strand, London, England, United Kingdom
¤i Current address: Environment and Sustainability Institute, University of Exeter, Penryn Campus, Penryn, Cornwall, United Kingdom
¤j Current address: Laila Nutraceuticals R&D Centre, JRD Tata Industrial Estate, Kanur, Vijayawada, Andhra Pradesh, India.
¤k Current address: Sygnature Discovery, BioCity, Nottingham United Kingdom
* s.d.c.griffin@leeds.ac.uk

**Data Availability Statement:** All relevant data are within the manuscript and its Supporting Information files.

## Abstract

Pandemic influenza A virus (IAV) remains a significant threat to global health. Preparedness relies primarily upon a single class of neuraminidase (NA) targeted antivirals, against which resistance is steadily growing. The M2 proton channel is an alternative clinically proven antiviral target, yet a near-ubiquitous S31N polymorphism in M2 evokes resistance to licensed adamantane drugs. Hence, inhibitors capable of targeting N31 containing M2 (M2-N31) are highly desirable. Rational *in silico* design and *in vitro* screens delineated compounds favouring either lumenal or peripheral M2 binding, yielding effective M2-N31 inhibitors in both cases. Hits included adamantanes as well as novel compounds, with some showing low

**Funding:** This work was supported by a Medical Research Council studentship awarded to CS (SG, RF), Medical Research Council grants G0700124 (S.G.) and L018578 (J.R.S), a Yorkshire Cancer Research Pump-Priming Award (S.G., LPP025), and a Pfizer Pump-Priming Award (SG). The funders had no role in study design, data collection and analysis, decision to publish, or preparation of the manuscript.

**Competing interests:** I have read the journal's policy and the authors of this manuscript have the following competing interests: Dr. Paul Targett-Adams is a former employee of Pfizer Ltd. He was involved collaboratively with this project during inception and early stages and acted as a contact and advocate during the application for the Pfizer pump-priming award. He is presently an employee of Sygnature Discovery, who are not involved with this work. C. Scott is a previous employee of ReViral Ltd. and is now an employee of Covance. However, neither body supplied funding to, nor influenced this manuscript.

micromolar potency versus pandemic "swine" H1N1 influenza (Eng195) in culture. Interestingly, a published adamantane-based M2-N31 inhibitor rapidly selected a resistant V27A polymorphism (M2-A27/N31), whereas this was not the case for non-adamantane compounds. Nevertheless, combinations of adamantanes and novel compounds achieved synergistic antiviral effects, and the latter synergised with the neuraminidase inhibitor (NAi), Zanamivir. Thus, site-directed drug combinations show potential to rejuvenate M2 as an antiviral target whilst reducing the risk of drug resistance.

## Author summary

"Swine flu" illustrated that the spread of influenza pandemics in the modern era is rapid, making antiviral drugs the best way of limiting disease. One proven influenza drug target is the M2 proton channel, which plays an essential role during virus entry. However, resistance against licensed drugs targeting this protein is now ubiquitous, largely due to an S31N change in the M2 sequence. Understandably, considerable effort has focused on developing M2-N31 inhibitors, yet this has been hampered by controversy surrounding two potential drug binding sites. Here, we show that both sites can in fact be targeted by new M2-N31 inhibitors, generating synergistic antiviral effects. Developing such drug combinations should improve patient outcomes and minimise the emergence of future drug resistance.

## Introduction

The 2009 H1N1 "swine 'flu" outbreak dramatically illustrated the speed at which influenza pandemics can spread in the modern era due to globalisation. Whilst not as virulent as the 1918 Spanish Influenza, which claimed more than 50 million lives, swine 'flu caused increased mortality and morbidity, placing considerable burden upon even advanced health care systems. The unexpected origin[1–3] of swine 'flu precluded the rapid deployment of a vaccine, making antiviral prophylaxis the only means by which to curtail the initial stages of the pandemic.

Neuraminidase inhibitors (NAi), including Oseltamivir ("Tamiflu", Roche), Zanamivir ("Relenza", GSK) and Peramivir ("Rapivab", BioCryst), dominate the current influenza antiviral repertoire[4–7]. Teratogenic effects limit use of a polymerase inhibitor licensed in Japan, Favipiravir[8, 9]. In 2018, the FDA approved the use of Baloxavir marboxil ("Xofluza", Roche/Shionogi) for uncomplicated influenza[10], which is a novel class of antiviral targeting cap-dependent endonuclease in both influenza A and B viruses. Worryingly, NAi resistance is increasing, with reports of mutant strains in the H7N9 highly pathogenic avian influenza (HPAI) epidemic in South East Asia[11–18]. Baloxavir marboxil resistance also occurs at a high rate (~10% of patients) [19] and Favipiravir selects resistance in pandemic H1N1 influenza cell culture[20].

Another class of influenza antiviral, the adamantane M2 proton channel inhibitors (M2i) amantadine and rimantadine, are now clinically obsolete due to widespread resistance[21, 22]. This is due to a near-ubiquitous S31N polymorphism within M2 (other rarer variants also occur) generating resistance at little or no associated fitness cost to the virus. Targeting rimantadine resistant M2 has been a long-standing priority yet progress targeting M2-N31 is limited

compared to other minor variants[23–31]. The majority of studies have focused upon adamantane derivatives with various chemical groups linked via the primary amine.

All influenza A strains require virion-associated M2 channels to mediate core protonation during endocytosis, promoting the uncoating of vRNPs following membrane fusion[32–34]. In addition, strains with a polybasic furin-cleavable haemagglutinin (HA) require M2 to prevent acidification of secretory vesicles to maintain glycoproteins in their pre-fusogenic state [35–41]. Mature M2 is a 96 aa type III membrane protein that forms tetrameric channels within membranes[42–45]. Acidic pH promotes M2 channel activity by both enhancing tetramer formation and the subsequent protonation of conserved His37 sensor residues within the channel lumen[42, 46–49]. This causes conformational shifts in adjacent Trp41 "gates" via a mechanism that remains debated[50–59]. More than twenty M2 atomic structures exist on the PDB, although none feature full-length protein. Instead, minimal "*trans*-membrane" (TM) or C-terminally extended "conductance domain" (CD) peptides have been investigated as these regions recapitulate channel function, although the CD region possesses enhanced biological activity[60]. Interestingly, drug-bound TM and CD structures differ with respect to adamantane binding[61, 62]; TM channels harbour a single lumenal amantadine molecule, whereas CD structures bind four rimantadine molecules at membrane-exposed peripheral sites, corresponding to the region largely absent from TM peptides. Uncertainty regarding amantadine/rimantadine binding remains, hampered by the poor chemical probe qualities of adamantanes and a lack of confirmatory functional studies comparing TM and CD peptides[63–73].

In the present study, we show that CD and TM peptides can be targeted by chemically-distinct small molecule inhibitors, suggesting that either both M2 binding sites, or potentially altered M2 conformations, are viable antiviral targets. Critically, this enables synergistic M2-targeted combination therapy. *In silico* high throughput screening enriched for novel compounds with predicted preference for one or other site, validated by the first comparative TM/CD peptide screen for M2-N31 channel activity. Several hits identified *in vitro* show antiviral activity versus pandemic H1N1 influenza A virus in the laboratory setting, comprising both modified adamantanes as well as unique scaffolds. Whilst a previously reported adamantane M2-N31 inhibitor rapidly selected resistance in culture, this did not occur for newly derived compounds. Excitingly, pairs of M2-N31 inhibitors achieved synergy, as did combining novel scaffolds with the NAi, Zanamivir. Together, these observations provide a firm basis for rejuvenating M2-N31 as a viable target for much needed drug combinations, which should help combat the emergence of antiviral resistance.

## Results

### Robust identification of specific M2-N31 inhibitors *in vitro*

We adapted an indirect liposome dye release assay for viroporin activity[74–78] for M2 CD region peptides derived from Influenza A/England/195/2009 (Eng195), a prototypical first wave virus from the 2009 H1N1 pandemic. In addition to the wild type Eng195 M2 harbouring N31, we included a mutated S31 peptide to allow validation with rimantadine (Fig 1A). Both peptides induced equivalent dose-dependent release of carboxyfluorescein (CF) from liposomes, and acidic pH increased M2 activity (S1A and S1B Fig). Critically, rimantadine only blocked activity of Eng195 M2-S31 peptides, confirming the ability of the assay to discriminate between susceptible and resistant M2 variants.

Modified adamantane compounds have been shown to inhibit M2-N31 activity[24, 26, 30, 31, 79]. Thus, to validate the assay we tested Eng195 peptides versus a small collection of similar prototypic molecules that included inhibitors of rimantadine-resistant hepatitis C virus (HCV) p7[80]. Encouragingly, from nine compounds tested, three modified adamantanes

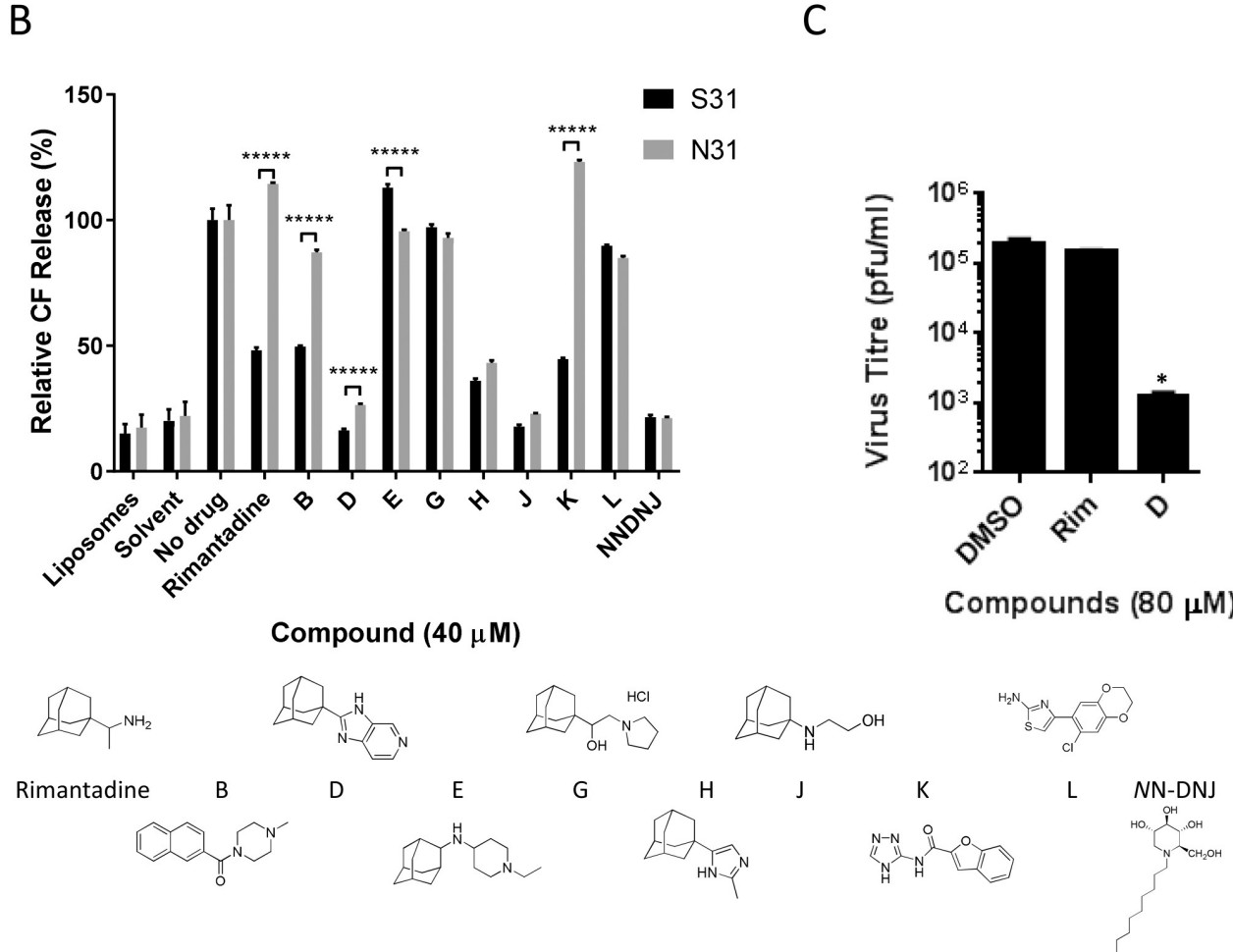

**Fig 1. Identification of M2-N31 targeted compounds from prototypic viroporin inhibitor classes. A.** Sequences of Eng195 M2 peptides (amino acids 18–60, corresponding to the "conductance domain" (CD)) used in study, comprising wild type (top) and N31S mutant (bottom). **B.** M2-N31/S31 containing Eng195 peptides were assessed for channel forming activity in DOPA/DOPC(DOPE) liposomes containing self-quenching concentrations of carboxyfluorescein. Activity relative to liposome-only, solvent and untreated M2 peptide controls (100%). At least two biological repeats were carried out, each with duplicate/triplicate technical repeats of each condition (***** $p \leq 0.00001$, paired student t-test). **C.** Cell culture antiviral effect of rimantadine and adamantane compound D upon Eng195 in a plaque reduction assay. Results are representative of three biological repeats, each containing two technical repeats (* $p \leq 0.05$, paired student t-test).

inhibited both M2-N31 and M2-S31 peptides (compounds D, H and J, Fig 1B). A further two adamantanes showed no activity (E, G), and another amiloride-related compound (L) was similarly inactive (Fig 1B). Interestingly, two amiloride-like molecules (B, K) showed activity versus M2-S31 but not N31, reminiscent of rimantadine. Lastly, the alkylated imino-sugar

*N*NDNJ also blocked the activity of both M2 peptides, yet this compound disrupted oligomerisation, as previously shown for HCV p7[80] (Fig 1A and S1C Fig). D, H, J and *N*NDNJ displayed antiviral effects in Eng195 culture (Fig 1C and S2 Fig), establishing the dye release assay as a robust means of screening for M2-N31 specific inhibitors with genuine antiviral effects.

## Ambiguous predicted binding modes for prototypic M2-N31 inhibitors

To gauge how novel M2-N31 inhibitors might bind the channel complex, we generated structural homology models for Eng195 M2-N31 and S31 based upon the PDB: 2LRF CD structure from the Chou laboratory[61] (Fig 2A and 2C). This template includes both potential rimantadine binding sites. As noted previously, N31 caused splaying of the *trans*-membrane domain (TMD) compared with the more lumenally oriented S31, yet the structure also differed throughout the helical bundle, consistent with a reported destabilising effect for N31[72] (Fig 2A and 2C). Surprisingly, docking of both rimantadine and novel inhibitors led to distinct binding poses at the lumenal and peripheral sites for N31 and S31 models. For the wild type Eng195 M2-N31 model, predicted binding at the peripheral site (defined by D44, R45 and F48) consistently involved H-bonding to D44, whereas the orientation of inhibitors altered in S31 models (Fig 2A, 2B and 2D, and S1 Table). Similarly, M2-N31 lumenal interactions occurred near to the N-terminal neck of the bundle near N31 and V27 (Fig 2A, 2B and 2D, and S1 Table), whilst binding within M2-S31 models resembled previous structures, occurring further inside the TMD just above H37. Such N/S31 dependent "flipping" within the channel lumen has been observed previously[30]. Based upon these observations, we reasoned that such promiscuous pleotropic binding may result from the chemical properties of adamantane derivatives, and that molecules with improved molecular fit may exhibit less ambiguous predicted binding modes. However, it would also be necessary to validate site preferences *in vitro* to generate meaningful structure-activity relationships (SAR) for improved M2-N31 inhibitors.

## Determination of M2-N31 inhibitor binding preference *in vitro*

TM peptides lack the majority of the C-terminal extension present within CD peptides that contains the proposed peripheral binding site. Thus, we hypothesised that lumenally targeted compounds would inhibit both TM and CD peptides, whereas those with a peripheral site preference would show activity only against CD peptides. The dye release assay was therefore adapted to include Eng195 TM peptides, accounting for their reduced biological activity compared to CD (Fig 3A and S1A Fig).

We first tested a published M2-N31 inhibitor, M2WJ332[81], a modified adamantane with activity against full-length M2-N31 shown to bind within the lumen of a TM domain NMR structure (A/Udorn/307/1972 (H3N2) M2-S31N, PDB: 2LY0, Fig 3B). Surprisingly, M2WJ332 blocked the activity of Eng195 M2-N31 CD peptides and had no TM-specific activity either under standard assay conditions (up to 40 μM) (Fig 3C). Thus, M2WJ332 displayed a strong functional preference for the peripheral binding site *in vitro* despite its location within the 2LY0 structure[81]. Accordingly, docking of M2WJ332 within the 2RLF-based Eng195 homology model resembled other adamantanes by generating poses within both the lumen and the peripheral site (Fig 3D and S4A Fig). To confirm this phenotype, M2-N31 peptide electrical properties were also tested using interface lipid bilayers in the presence/absence of M2WJ332 (Fig 3E and 3F, and S3 Fig). Reassuringly, M2WJ332 had remarkably more potent effects verses CD, rather than TM peptides, confirming that potential artefacts from dye release assays were not responsible for erroneous M2WJ332 specificity; moreover, TM peptides remained susceptible to other inhibitors (see below and S3E Fig). Whilst structural and biophysical

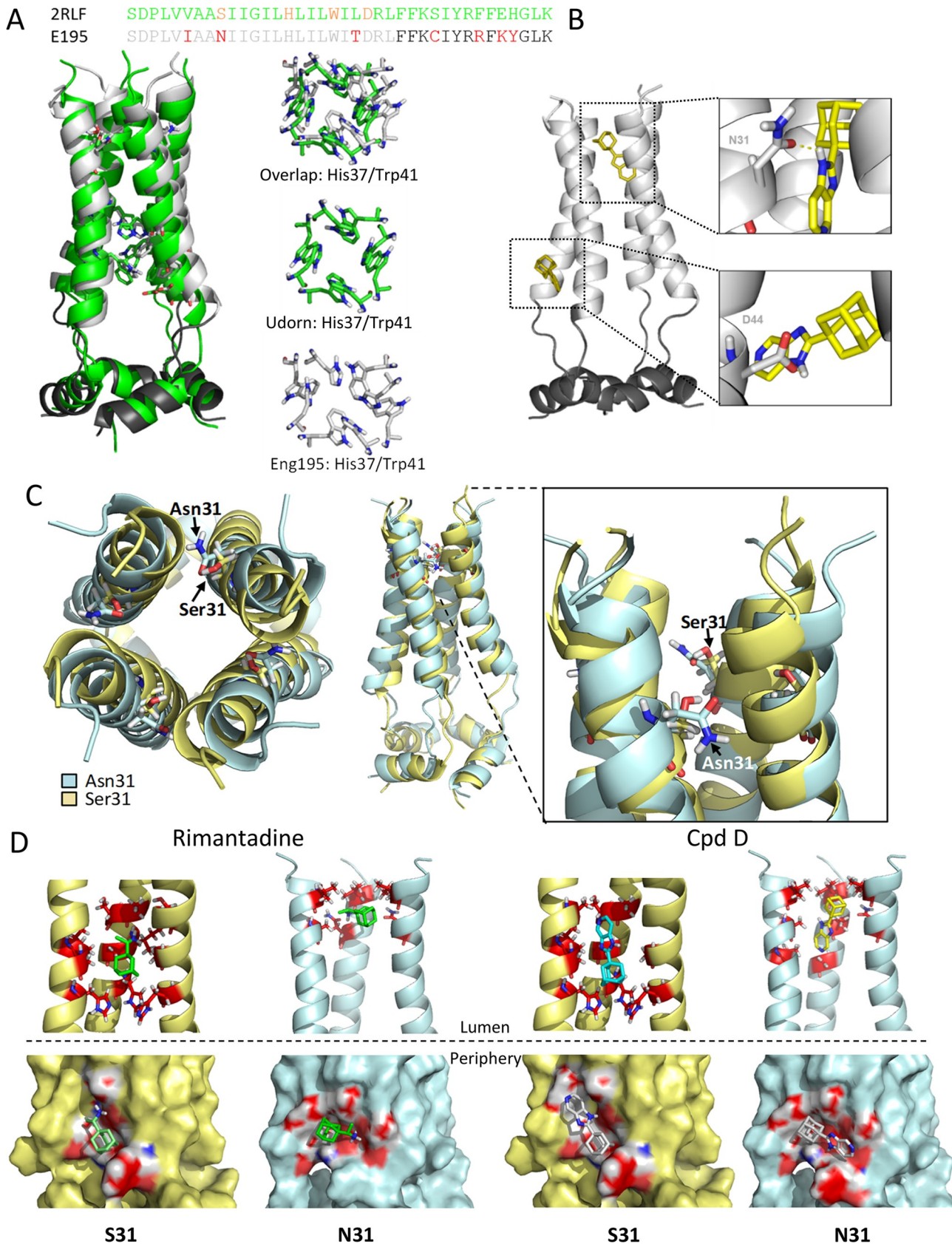

**Fig 2. Eng195 homology models based upon PDB: 2RLF conductance domain structure. A.** Overlapping structures of original 2RLF structure (green) and Eng195 (E195) homology model (light/dark grey corresponding to TM and C-terminal extension of CD, respectively). **B.** Docking of compound D into lumenal and peripheral binding site within Eng195 model. **C.** Alteration of Eng195 structure caused by occupation of position 31 by either serine (gold) or asparagine (pale blue). Left: top-down view showing lumenal orientation of Ser31 versus inter-helical Asn31. Middle and zoom: side view of helical displacement and location of Ser31 versus Asn31 sidechains. **D.** Docking of Rimantadine and compound D into lumenal (top) or peripheral (bottom) binding sites of Eng195 homology model with either Ser31 (gold) or Asn31 (pale blue).

studies have previously compared lumenal and peripheral binding[67, 69], to our knowledge this represents the first functional evidence supporting the relevance of peripherally targeted M2 ligands *in vitro*. Importantly, this suggests that M2-N31 possesses two potential binding sites to exploit for antiviral discovery.

## Screening of novel M2-N31 inhibitors enriched for lumenal or peripheral site preferences

The next step was to enrich *in silico* screening libraries for compounds predicted to bind preferentially at one or other M2 site, removing as much ambiguity as possible through extensive attrition of compound characteristics. Using the Eng195 2RLF-based model as a template, grids corresponding to each site were targeted by an *in silico* high throughout screen (eHiTS, SimBioSys Inc.), based upon a random chemical library and a second input ligand pool derived through evolution of the compound D molecular structure (ROCS, OpenEye Scientific) (Fig 4A). eHITS score ranked the top 1000 hits for each site and docking scores were cross-validated using a second software package (SPROUT, Keymodule Ltd.). Short-listing of compounds involved an attrition protocol directed by agreement between the two binding scores, compound molecular weight, specific binding pose and drug-like qualities. Prioritisation of compounds focused upon site selectivity, rather than merely predicted potency. Details of resultant compounds are summarised in S2 Table.

Compound screens for activity versus TM and CD peptides at 40 μM yielded multiple hits (defined as a ≥30% reduction in channel activity for at least one M2 peptide at 40 μM) corresponding to lumenal and peripheral site preferences. Interestingly, more lumenally targeted hits presented compared to peripheral, and a third class of compound displayed specificity to TM rather than CD peptides, e.g. compound P6.4 (S2 Table). A minority of compounds displayed functional site preferences contradicting docking predictions, yet rational enrichment of ligand pools *in silico* had significantly augmented the number of M2-N31 targeted hits, with ~50% of compounds displaying M2-inhibitory activity *in vitro* compared with a hit rate of <1% from a random prospective screen[82].

## Lumenally and peripherally targeted M2-N31 inhibitors exert antiviral effects against Eng195 in culture

We next titrated exemplar compounds from each class to ensure specificity corresponded to that observed in the 40 μM screen and predicted interactions (Fig 4B and S4B Fig). Interestingly, whilst lumenal compounds (e.g. L1.1) displayed equivalent activity against both TM and CD peptides, some peripherally targeted ligands (e.g. DP9) also began to exert measurable effects versus TM peptides at higher concentrations. This included DL7, which despite predictions of lumenal binding, displayed clear preference for CD peptides at lower concentrations (Fig 4B). We hypothesise that this occurs due to inefficient interactions with the partial peripheral binding site present at the C-terminus of TM peptides. Moreover, titrations confirmed the phenotype of TM-specific ligands (e.g. P6.4) (Fig 4B).

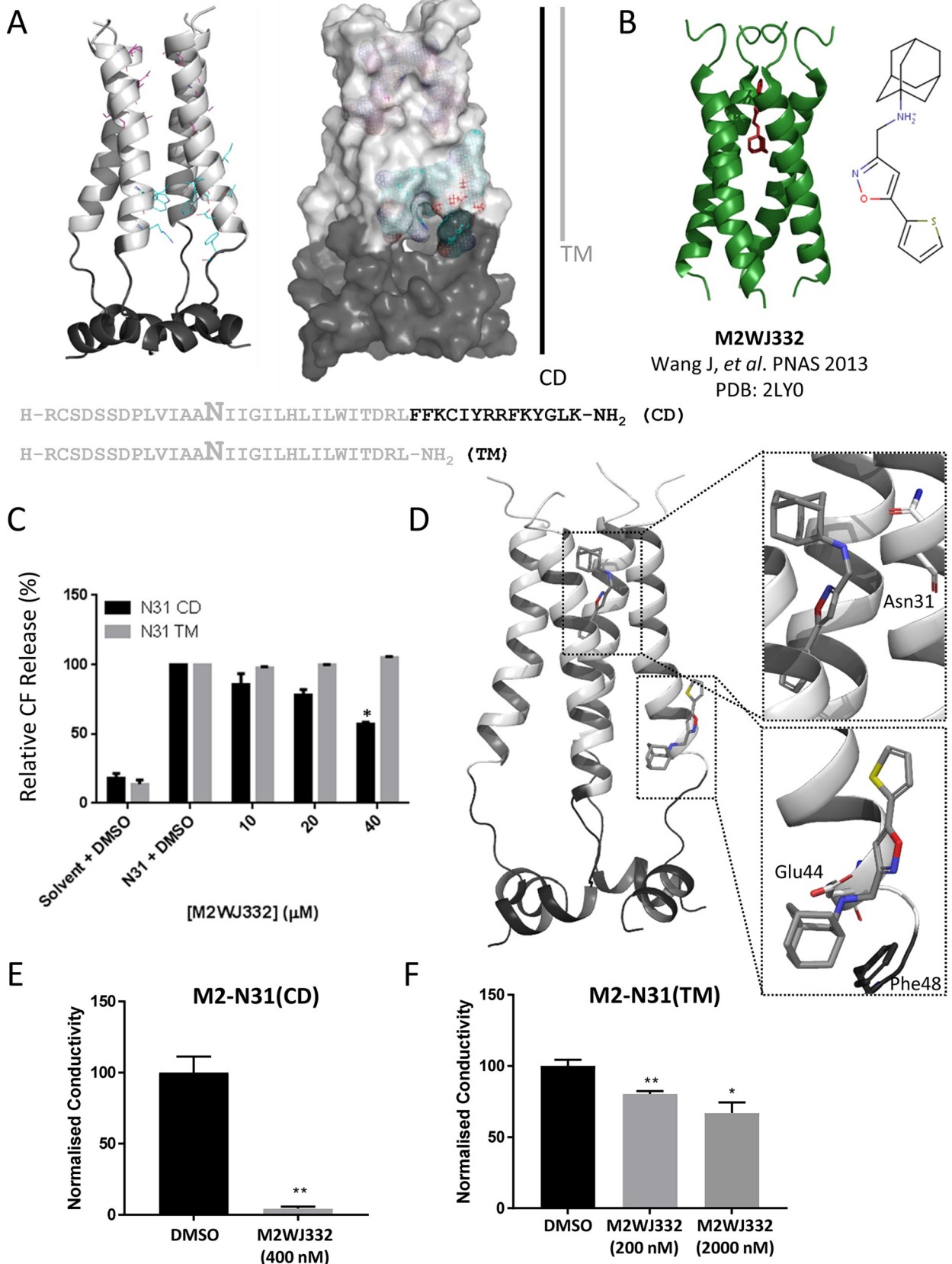

H–RCSDSSDPLVIAA**N**IIGILHLILWITDRL**FFKCIYRRFKYGLK-NH₂** (CD)

H–RCSDSSDPLVIAA**N**IIGILHLILWITDRL-NH₂ **(TM)**

**M2WJ332**
Wang J, *et al*. PNAS 2013
PDB: 2LY0

**Fig 3. Analysis of adamantane M2WJ332 biological activity versus M2-N31. A.** Structure, sequence composition and binding site grids within M2-N31 TM (light grey) and CD (C-terminal extension in dark grey) peptides. **B.** PDB 2LY0 adapted from Wang *et al.*, PNAS 2009 showing M2WJ332-bound M2 TM domains from an S31N-mutated H3N2 M2 channel complex (Udorn). **C.** Activity of M2WJ332 in liposome dye release assay versus M2-N31 TM and CD peptides. Data representative of multiple biological repeats, each containing two technical repeats (* $p \leq 0.05$, paired student t-test). **D.** Predicted docking poses for M2WJ332 at both lumenal and peripheral binding sites upon the Eng195 2RLF-based channel complex homology model, generated using e-Hits (see also S4A Fig for predicted interactions). **E.** Inhibition of M2-N31 CD peptide channels using M2WJ332 within interface lipid bilayers. Results from three independent experiments. Error bars represent standard error of the mean with $p$ values determined using the student T-test (* $p \leq 0.05$, ** $p \leq 0.01$). **F.** As for **E** using M2-N31 TM peptides.

Following cytotoxicity testing in MDCK culture (S5 Fig), hit compounds were screened for antiviral effects in Eng195 plaque reduction assays at 80 μM (Fig 5A). Lumenally targeted L1.1 was the most potent compound, achieving almost a 3 $\log_{10}$ reduction in virus titre. Other hit compounds, as well as M2WJ332, achieved a 1–2 $\log_{10}$ reduction. Interestingly, with the exception of a modest effect for P6.4, none of the compounds showing TM peptide specific preferences *in vitro* displayed antiviral effects against Eng195 in culture. The most potent lumenal (L1.1) and peripheral (DP9) hits, along with M2WJ332 (peripheral) and DL7 (peripheral/lumenal?), advanced to further studies. With the exception of DP9 ($25 < IC_{50} < 50$ μM), shortlisted compounds displayed cell culture $IC_{50}$ values between 1–2 μM (Fig 5B). Thus, data supported that rational enrichment targeting both distinct binding sites had successfully yielded M2-N31 inhibitors with potent antiviral effects.

## Evolution of Eng195 in M2-N31 inhibitor chronic culture

It was important to establish whether the low genetic barrier to rimantadine resistance was also relevant to M2-N31 inhibitors. Serial Eng195 MDCK supernatant passage (Fig 6A) was conducted with increasing concentrations of M2WJ332, L1.1 and DL7 (DP9 excluded as precise $IC_{50}$ not determined). Eng195 replication was monitored by periodic titration and Sanger sequencing of M2 RNA in supernatants. The only change in the M2 sequence was detected starting from the first analysis of M2WJ332-selected supernatants (day 5); a U>C change at position 80 (M2 cDNA sequence) led to a Val27>Ala mutation (GUC>GCC, V27A) (Fig 6B). This polymorphism enriched over time, becoming the dominant species apparent at day 14 (Fig 6B).

Sequencing of plaque-purified virus from day 5 supernatants confirmed the presence of V27A within 7/7 M2WJ332-selected plaques, whereas 9/9 L1.1-, and 5/5 rimantadine-selected plaques retained wild type M2-N31 sequence (Fig 6B). Accordingly, far fewer plaques were derived from normalised L1.1 supernatants compared to M2WJ332 or rimantadine (S6 Fig), and these were eliminated by limited titration of the compound. By contrast, M2WJ332-selected supernatants still retained multiple plaques (~30% of DMSO control) at much higher concentrations (80 μM), likely reflecting the proportion of mutant virus (~30–40%, see below) within the bulk population. Lastly, virus could only be expanded from L1.1 plaques at ≤20 μM inhibitor, with cytopathic effects (CPE) taking at least 48 h to manifest. By contrast, M2WJ332 or rimantadine plaques readily expanded under 80 μM inhibitor, with CPE evident by 24 h.

To investigate further whether V27A was potentially linked to M2WJ332 resistance, the fold increase in titre was determined under selection (80 μM compound) for passages six and seven. Eng195 under Rimantadine, Zanamivir (known to rapidly select resistance[83, 84]) and M2WJ332 achieved similar fold-increases compared to DMSO controls, whereas both L1.1 and DL7 significantly suppressed viral replication leading to much reduced titres compared to input (Fig 6C). We then introduced an evolutionary bottleneck at passage eight to enrich for any minor resistant variants present within bulk populations, normalising innoculae to a multiplicity of infection (MOI) of 0.001. After a further six passages, output titres (passage 14)

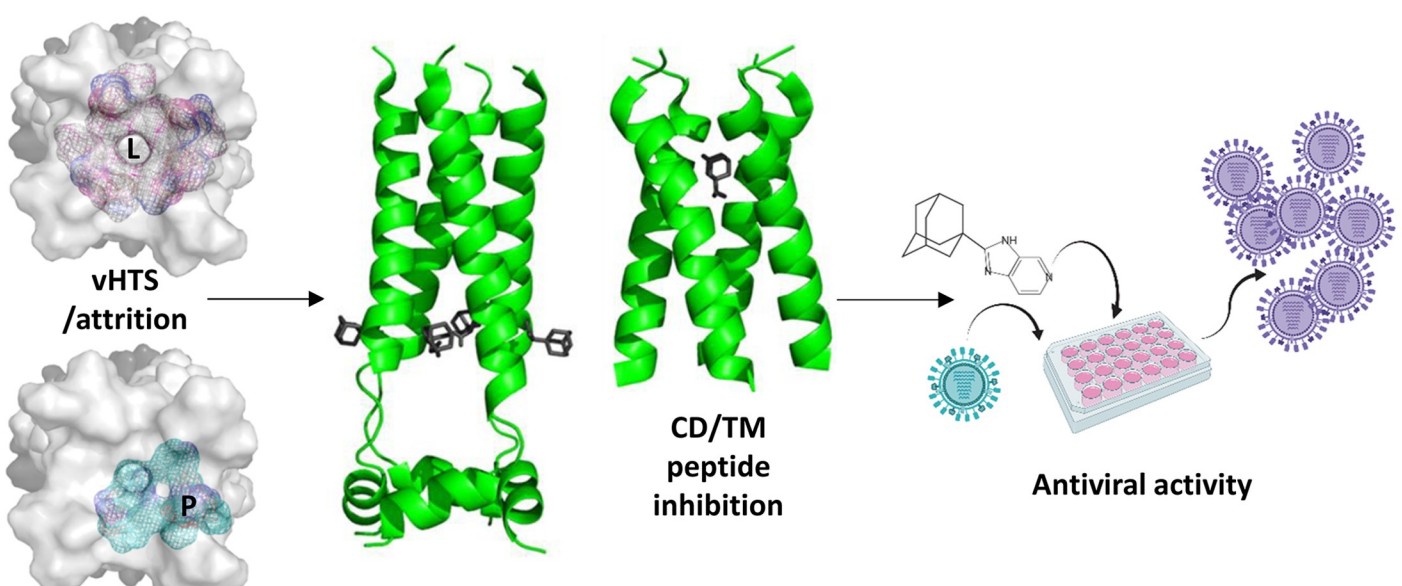

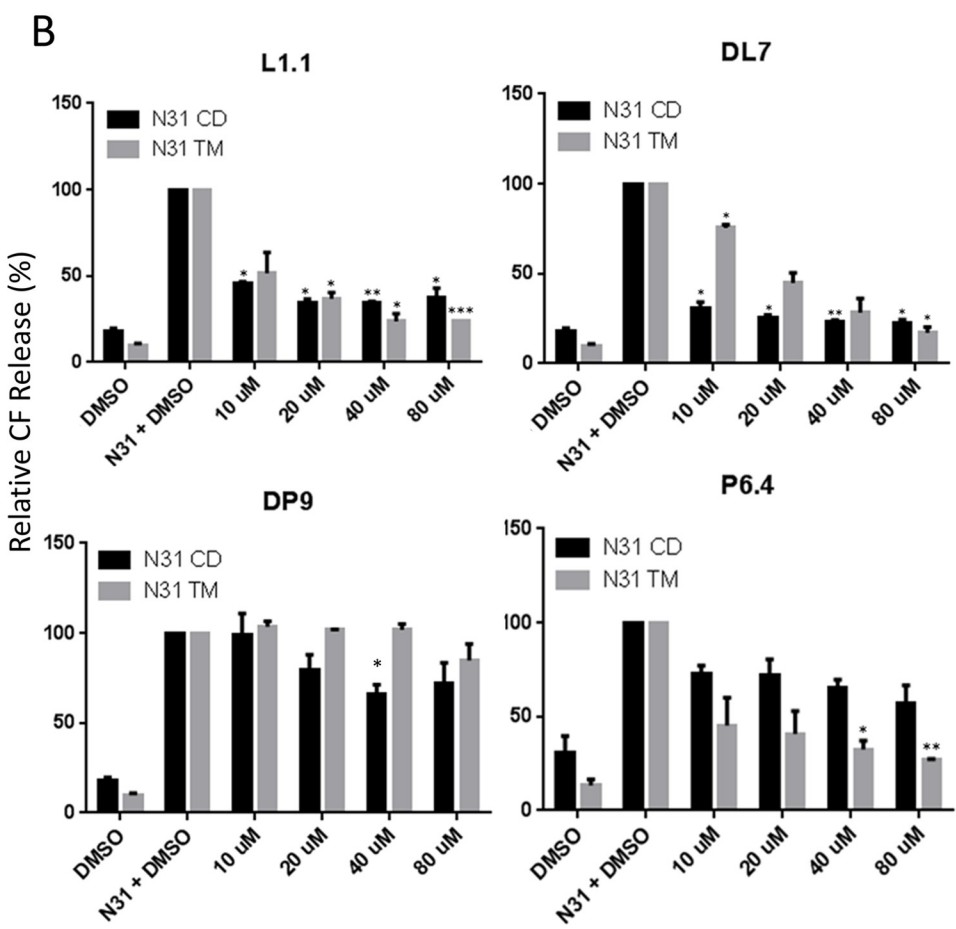

**Fig 4. Selection and characterisation of lumenally or peripherally targeted M2-N31 ligands. A.** Schematic of inhibitor assay pipeline: left—*in silico* compound selection via vHTS showing respective target grids for lumenal and peripheral binding sites (top-down view of M2-N31 tetramer), middle—validation of binding versus TM and CD Eng195 peptides, right–inhibition assays in Eng195 MDCK culture (Imagery designed using "Biorender" https://app.biorender.com). **B.** Titration of class exemplar compounds versus M2-N31 TM/CD peptides in liposome dye release assays: Lumenally targeted L1.1 showing equivalent activity versus both TM/CD; peripherally targeted DP9 showing preferential inhibition of CD peptides; DL7 showing preference for CD peptides but with activity versus TM at higher concentrations; P6.4 showing preference for TM peptides (* p≤0.05, ** p≤0.01, *** p≤0.001, paired student t-test).

again revealed a significant reduction in L1.1 selected Eng195 titre, whereas DL7 selected supernatants had recovered to a similar range as controls (Fig 6D).

Finally, deep sequencing of IAV genomes was performed comparing passage 5 and 14 supernatants to investigate minor M2 variant populations and mutations occurring elsewhere in the Eng195 genome (Fig 6E). Only M2WJ332 selected virus showed changes in the M2-N31 sequence compared to controls, with V27A increasing from ~40 to ~80% abundance between the two time points. Additional minor changes also occurred at position 31 (N31S/I), with another change located outside of the CD region (E70K). Low prevalence changes also occurred in the HA protein in M2WJ332 selected virus at passage 14, namely K226E, Y454H/S and N461D. L1.1 selected virus also showed a low prevalence change at HA Y454H, along with changes in PB2 D161N and M1 P55S. No variation in M2 sequence was evident for DL7 selected virus at passage 5 or 14, making it difficult to understand the mechanism by which titre had initially been suppressed and then subsequently rebounded. Interestingly, unlike L1.1 and M2WJ332, DL7 selection drove only two non-synonymous changes within the viral genome, one each at passage 5 and 14, even though the diversity of synonymous changes was similar across all conditions, including DMSO controls (Table 1, S3 Table). Hence, it appeared that DL7 was not in fact driving the evolution of viral sequences over multiple passages, consistent with this compound in fact being inactive during prolonged cell culture. Taken together, V27A was the only relevant polymorphism significantly enriched during chronic culture with novel M2-N31 inhibitors. This strongly suggests that Eng195 M2-A27/N31 confers resistance to M2WJ332.

## Synergistic antiviral effects using M2-N31 targeted inhibitor combinations

M2-N31 specific inhibitors with distinct binding properties provides the opportunity for antiviral combinations that could not only improve therapy, but also reduce the likelihood of resistance. Combinations of M2WJ332, DP9, DL7 and L1.1 were titrated in Eng195 MDCK plaque reduction assays and antiviral effects assessed using MacSynergy software (results corroborated using "Compusyn"). Remarkably, combinations of M2WJ332 with either L1.1 or DP9 yielded synergistic reductions of viral titre (Fig 7A). M2WJ332 combined with L1.1 showed increased synergy proportionate to both inhibitor concentrations. However, synergy between M2WJ332 with DP9 only occurred in the lower M2WJ332 range and increased with DP9 concentration. By contrast, combinations involving the DL7 compound resulted in antagonism, whether combined with a lumenally (L1.1) or peripherally (M2WJ332) targeted partner (Fig 7B); this may relate to the observed lack of antiviral effect in long-term culture (see above). Lastly, L1.1 also achieved synergistic antiviral effects when combined with the NAi, Zanamivir (Fig 7C and S7 Fig), supporting that drug combinations between classes should be achievable.

## Discussion

This work lays the foundation for future combination therapies targeting rimantadine-resistant influenza A viruses, which could form a vital addition to the current pandemic antiviral repertoire. We have moved beyond the debate surrounding two potential binding sites within

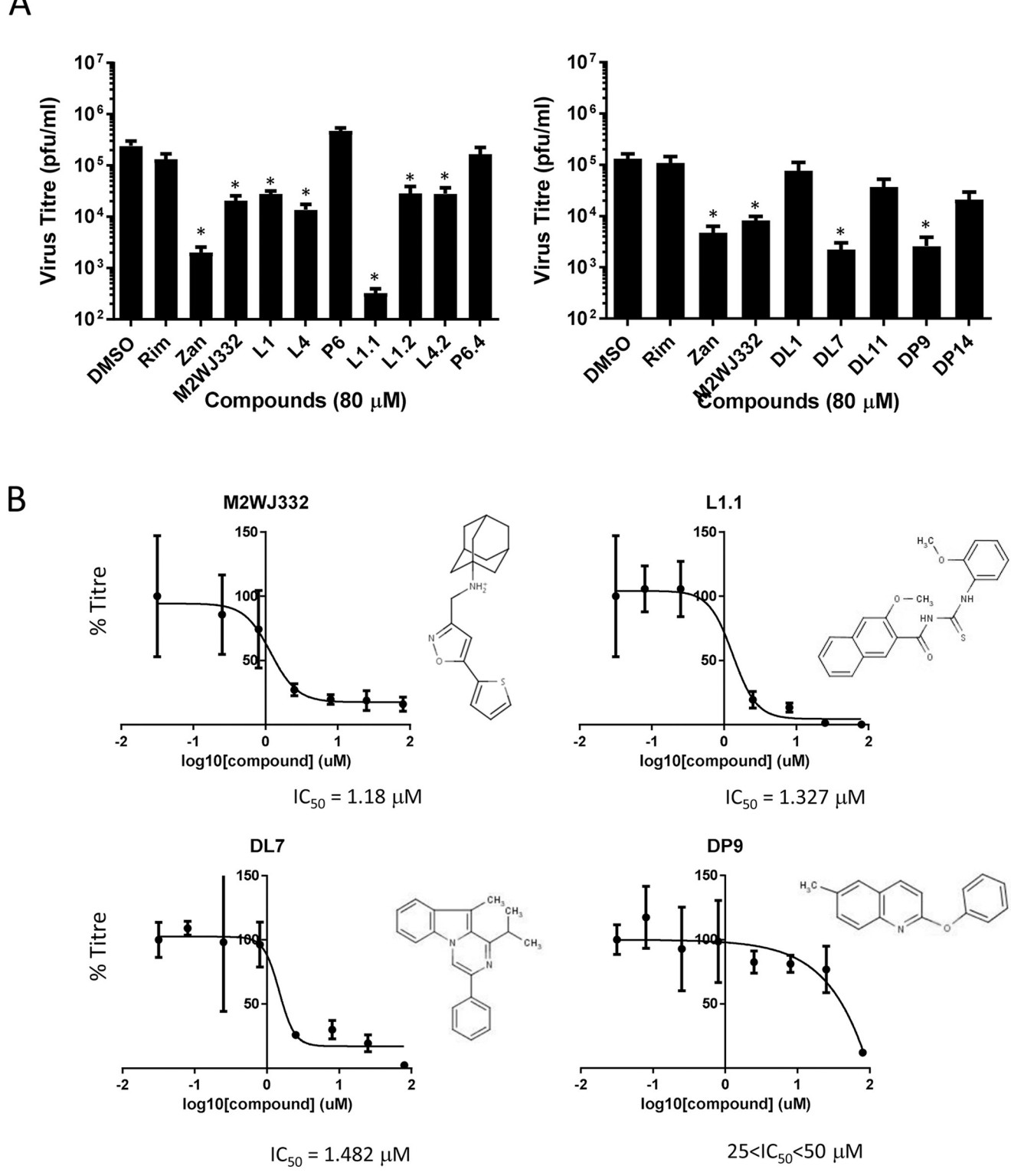

**Fig 5. Activity of compounds showing specific M2-N31 activity versus Eng195 in vitro in MDCK plaque reduction assays. A.** Compounds present at 80 μM both during infection (MOI of 0.01 pfu/cell) and through a 24 hr incubation in producer cells prior to titration of secreted infectivity. Rimantadine (80 μM) negative and Zanamivir (20 μM) positive controls were included. Data are representative of at least three biological repeats containing duplicate technical repeats (* p≤0.05, paired student t-test). **B.** 8-point $IC_{50}$ calculations across $\log_2$ concentration steps (triplicates for each point) for hit compounds L1.1, DL7 and DP9, alongside M2WJ332 in MDCK plaque reduction assays, shown with corresponding molecular structures.

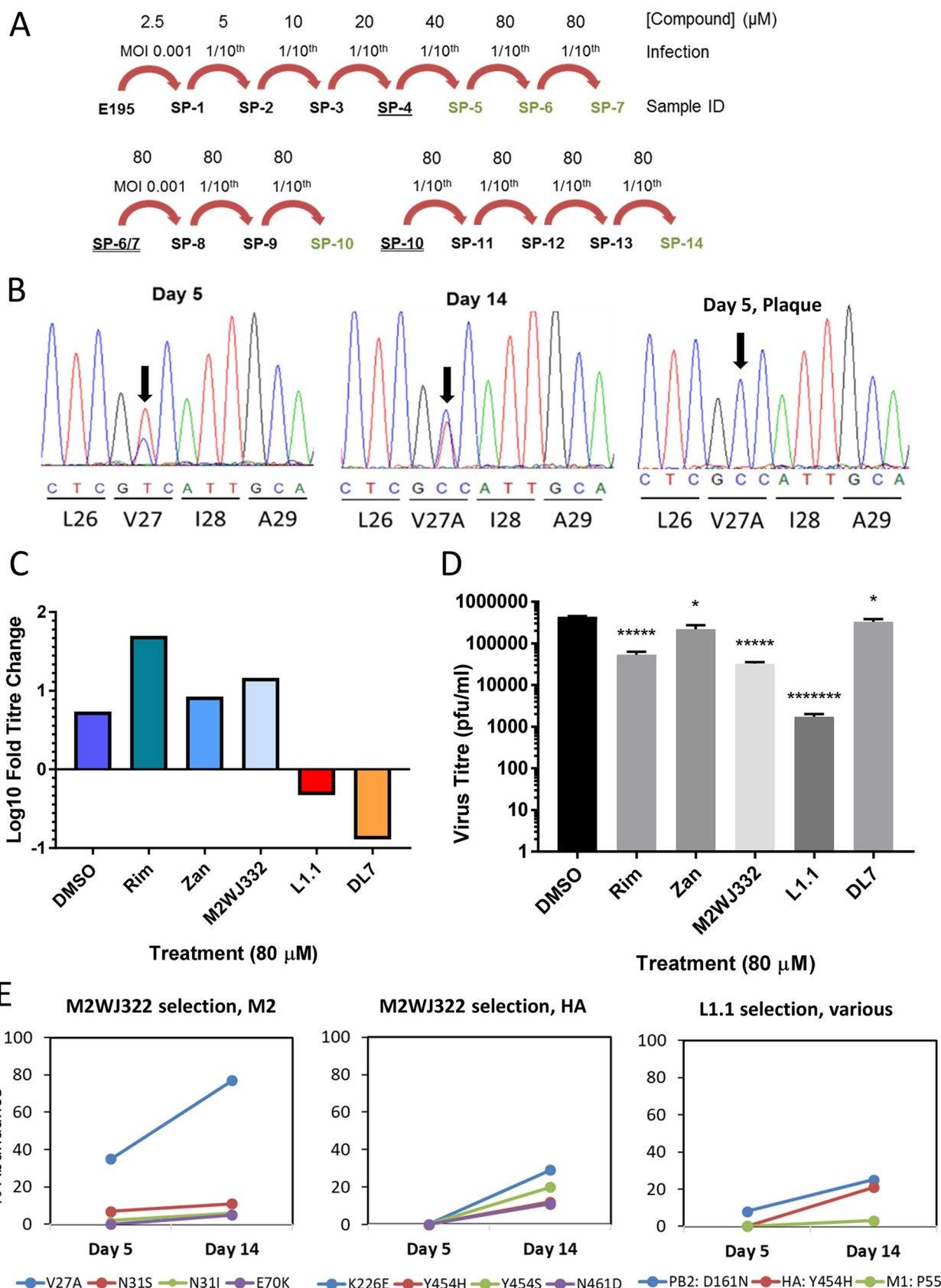

**Fig 6. Evolution of Eng195 in culture with targeted compounds. A.** Schematic of serial supernatant passage of Eng195 with increasing inhibitor concentrations. Underlined Sample IDs (serial passage, "SP"1, 2, 3 etc) were stored at 4 ºC for at least 24 hr prior use. Double underlined were clarified, snap frozen and then thawed prior to use. Green text indicates that the M2 protein was sequenced (bulk direct sequencing). **B.** Direct sequencing of secreted vRNA throughout serial passages of M2WJ332-treated virus showing passage 5 and 14 bulk populations, as well as an example of plaque purified virus from day 5. Progressive enrichment of V27A is apparent and becomes absolute in the plaque-purified samples. **C.** Monitoring of viral suppression during the course of selection. Quantification of SP7 output titres after infection by equivalent doses of SP6 innoculae. **D.** SP14 titres following imposition of bottleneck at SP8 (* p≤0.05, ***** p≤0.005. ******* p≤0.000005, paired student t-test). **E.** Longitudinal quantitative whole genome deep sequencing analysis for virus treated with M2WJ332, L1.1 or DL7. Percentage prevalence of changes relative to controls are shown for SP5 and SP14. M2WJ332 selects both V27A and other minor variants within M2, yet no such changes occur in L1.1 treated cultures. Other minor variants are selected over the course of the experiment in HA and other ORFs for both compounds.

M2 channel complexes, instead showing that synergistic antiviral therapy is achievable using compounds designed to target one or other region, which ultimately should reduce the incidence of new resistance mutations. Finally, whilst adamantanes still contribute to the M2-N31 chemical toolbox, we describe multiple distinct scaffolds that should provide a start-point for the next steps in antiviral drug discovery.

The question of how amantadine and/or rimantadine block the activity of M2 from sensitive influenza strains has been debated since two contrasting atomic structures were published in 2008[61, 62]. However, these and other ensuing studies generally compared TM with CD peptides, which is likely to bias where prototypic adamantanes bind. This is due to the C-terminal extension in CD peptides inducing a more compact helical bundle that is less favourable for lumenal interactions compared with the much broader structure seen for TM peptides, which also lack the majority of the peripheral binding site[67]. Moreover, it is increasingly clear that M2 peptides are highly malleable, with variable solubilisation conditions exerting profound effects upon structural studies[50].

Numerous biophysical and other investigations have attempted to define the principle amantadine/rimantadine binding site, including comparative NMR- and SPR- based binding studies[67, 69]; current consensus favours the lumen, but with secondary interactions at the periphery, although many such studies do not include CD peptides. However, perhaps the most relevant M2 structure solved for lipid-borne CD peptides (PDB: 2L0J)[85] rarely features in biophysical or other studies, likely due to the technical difficulties associated with the use of membrane bilayers compared with membrane-mimetic detergents. Notably, the lipidic 2L0J membrane bundle is less compact compared with other CD structures, and the orientation of the C-terminal extension, comprising basic helices that form the core of the peripheral binding site, differs significantly in 2LOJ compared to the detergent-solubilised template used for the present study, 2RLF[61, 85]. This may explain why fewer peripherally targeted compounds were selected compared to the lumen; accordingly, re-docking peripheral compounds into

**Table 1. Synonymous (syn) and non-synonymous (non-syn/NS) changes in E195 whole genome sequencing over the course of fourteen passages of drug selection.**
Virus RNA derived from supernatant passages five and fourteen was purified and sequenced, prior to the evolutionary bottleneck and the final passage, as well as a plaque-purified sample virus from passage five. Data reflects the number of discrete changes present within the sample rather than their prevalence within the population. Complete list of changes is provided in S3 Table.

| | Day 5 | | | Day 14 | | | | |
|---|---|---|---|---|---|---|---|---|
| Condition | Syn | Non-Syn | Total | Syn | Non-Syn | Total | Total Changes | % NS |
| DMSO | 36 | 3 | 39 | 38 | 2 | 40 | 79 | 6.32911392 |
| M2WJ332 | 10 | 3 | 13 | 45 | 12 | 57 | 70 | 21.4285714 |
| M2WJ332 plaque | 11 | 3 | 14 | ND | ND | | 14 | 21.4285714 |
| L1.1 | 22 | 3 | 25 | 31 | 9 | 40 | 56 | 21.4285714 |
| L1.1 plaque | 16 | 3 | 19 | ND | ND | | 19 | 15.7894737 |
| DL7 | 29 | 1 | 30 | 32 | 1 | 33 | 63 | 3.17460317 |

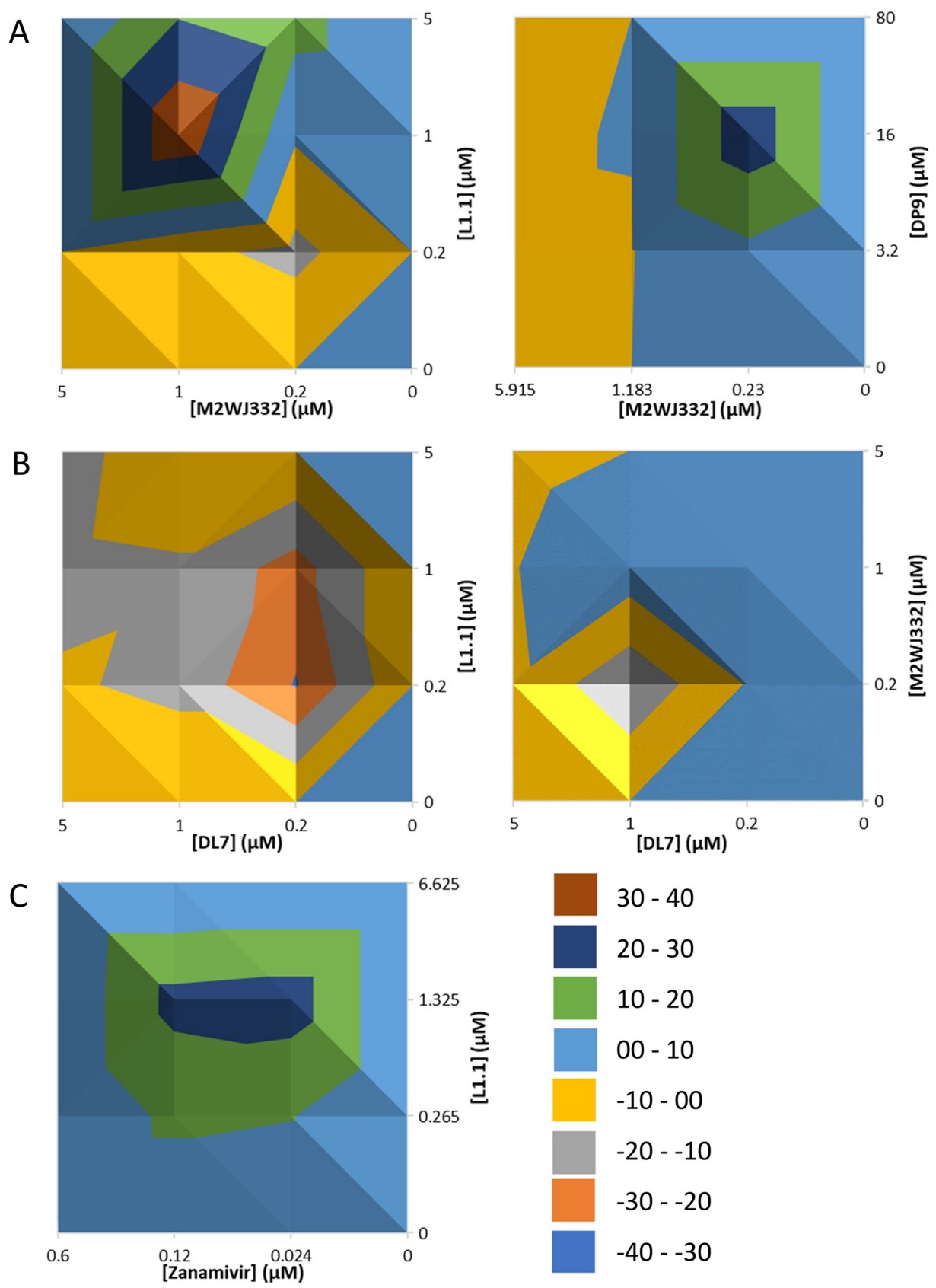

**Fig 7. Antiviral effects of M2-N31 inhibitor combinations versus Eng195 MDCK culture.** Class exemplar compounds were used in combination regimens across matrices of increasing concentrations. Resultant antiviral effects were assessed, and MacSynergy used to determine potential synergistic/additive/antagonistic effects. **A.** Top-down views of 3D-relief plots are shown with coloured peaks/troughs representative of synergistic (i.e. >0) or **B.** antagonistic (i.e. <0) effects. **C.** Synergistic MacSynergy plot of M2-N31 inhibitor L1.1 used in combination with Zanamivir (NAi).

2LOJ using E-Hits results in altered binding poses and affinity scores (S8 Fig). It would be of great interest to repeat our *in silico* enrichment using this template to determine whether the same or altered hits ensued, potentially with greater numbers of peripherally targeted compounds.

To our knowledge, the present study is the first to compare TM and CD peptides using an *in vitro* functional assay. Our hypothesis assumes that the presence/absence of the C-terminal extension discriminates peripheral binding, thus the identification of CD peptide-specific hits may represent direct functional evidence that the peripheral binding site represents a druggable target for M2. However, it is also possible that compound specificity may be affected by altered conformations adopted by CD and TM peptides within liposomes. The equivalent effects seen for lumenally targeted compounds suggests that the *trans*-membrane region is unlikely to differ between longer and shorter peptides, yet whilst the presence/absence of the C-terminal region is the most obvious explanation for CD-specific compounds, this is still not definitive. Interestingly, a third class of compounds was identified, including P6.4, that showed *in vitro* preference for TM peptides, suggesting that another region of TM from that targeted by lumenal binders can adopt a distinct structural conformation compared to CD peptides. However, with the exception of a mild antiviral effect for P6.4 (Fig 5A and S2 Table), none of these compounds displayed activity in Eng195 culture making their biological relevance unclear.

Other M2-N31 targeted studies, such as that describing M2WJ332[81], combine TM-derived structures with compound efficacy data from full-length protein, thereby presuming that the two structures are directly related. Nevertheless, discovering that M2WJ332 showed a functional preference for the peripheral binding site was unexpected. It is conceivable that M2WJ332 interacts differently with Eng195 M2 compared to the protein in the reported TM structure (A/Udorn/307/1972 (H3N2) M2-S31N, PDB: 2LY0). Moreover, 2LY0 was solved using relatively harsh detergent conditions[81, 86], rather than lipid, and in the presence of millimolar rather than micromolar inhibitor concentrations. Nonetheless, whether or not Eng195 and Udorn M2 are directly comparable as proteins, the present study serves as precedent for effective influenza A virus inhibitors targeting either the M2-N31 peripheral site, or a conformation distinct from that favouring lumenal binding, for at least one influenza A virus strain, raising the tantalising prospect of M2-targeted drug combinations.

To exclude potential artefacts from the use of indirect dye release assays when determining binding site preferences, we also examined CD and TM peptides in a supported bilayer electrophysiology assay (S3 Fig). Reassuringly, CD peptides were sensitive to nanomolar concentrations of M2WJ332, whereas TM peptides retained channel activity in the presence of much higher concentrations; L1.1 still inhibited TM peptides (Figs 3E, 3F and S3E). M2 peptides exhibited single channel activity in this system, yielding currents in the low pico-Ampere (pA) range as seen in other bilayer studies using electrochemical gradients based upon potassium ions [43]. However, these properties differ from behaviour of M2 expressed in mammalian cells when conducting protons, which are likely to be the biologically relevant substrate, where channel conductance was estimated to comprise femto-Ampere (fA) currents [87]. Other virus-encoded ion channels have also been shown to transport multiple ionic species and this may, or may not relate to biologically relevant behaviour *in vitro* and what causes these

differences in conductance is unclear. Moreover, similar questions arise around using CF as a proxy measure of M2 activity within screening assays, and this may relate to the relatively high concentrations of inhibitor required to block channel activity compared to full-length protein in *Xenopus* Oocytes, for example. Nevertheless, M2 peptide activity was influenced by acidic pH, compound specificity was confirmed by rimantadine sensitivity dependent upon the S/ N31 resistance polymorphism, and significant enrichment of biologically active inhibitory compounds was achieved.

Eng195 chronic culture in the presence of M2WJ332 led to the rapid evolution of a V27A change within the M2 sequence. Both plaque purification and monitoring the titre of selected bulk populations supported that this change confers specific resistance. V27A is a known amantadine resistance mutation[88, 89], albeit less prevalent compared to S31N. Amantadine binds proximal to H37 in the central portion of the *trans*-membrane helical bundle[64], meaning that S31N and V27A are too distant for these mutations to affect direct contacts with the drug, making the mechanism of resistance unclear. However, direct contact with V27 and N31 is predicted for lumenally docked compound D and M2WJ332 in the E195 homology model (Figs 2D and 3A), and this is seen in the 2LY0 TM structure where M2WJ332 interacts with the lumen [81]. Thus, an obvious explanation for the mechanism of M2WJ332 resistance would be the disruption of this interaction and resultant destabilisation of compound binding, contradicting the functional preference for M2WJ332 identified herein. However, the 2LY0 structure lacks the peripheral binding site and M2WJ332-bound CD peptide structures are not available; as discussed, both peptides also confer distinct structural bias. A distinct allosteric mechanism must therefore exist in order for V27A mutations to influence peripheral site interactions with M2WJ332. Interestingly, both S31N and V27A mutations destabilise CD peptide M2 channel complexes, the effects of which were proposed to disrupt interactions with peripherally bound rimantadine[72]. V27 is also proposed to form a secondary gate/constriction at the neck of the channel lumen[90], and a CD structure containing this mutation had both a wider lumen as well as an altered conformation in the peripheral binding region [91]. Thus, it seems that the M2 lumen and peripheral regions are structurally linked, meaning that mutations in one region might be expected to influence the other. In this regard, resistance to a compound related to M2WJ332 was shown recently to arise via a L46P mutation, i.e. located within the peripheral binding pocket [23]. Hence, it seems neither possible to reinforce nor contest *in vitro* data on M2WJ332 peripheral binding based upon the location of V27A at present.

Notably, naturally occurring M2-N31/A27 double variant isolates exist, implying a low genetic barrier in nature as well as in cell culture[92]. Interestingly, other minor M2 variants selected by M3WJ332 included the revertant N31S, which mediates resistance to another published M2-N31 specific adamantane derivative; a more dramatic N31D mutation also mediated resistance to a dual M2-N31/S31 inhibitor[93]. However, changes in M2, including at N31, did not occur in L1.1-selected Eng195, which forms predicted interactions with N31 but not V27 due to sitting lower in the lumen interacting with the H37 tetrad (S4 Fig). Accordingly, this compound maintained suppression of Eng195 bulk titre throughout the course of the experiment. DL7 appeared to behave similarly to L1.1 at early times, but titres recovered following the introduction of the evolutionary bottleneck at passage eight. A lack of non-synonymous sequence changes relative to controls or other drug treatments makes it unlikely that the compound was driving evolution of the virus in the same way that both L1.1 and M2WJ332 did and we conclude that this compound was not exerting an antiviral affect over the course of the resistance experiments. However, this contradicts the action of the compound versus peptides *in vitro* as well as in 24 hr plaque reduction assays. We speculate that DL7, like several other compounds we identified, exerts M2 inhibitory activity *in vitro*, but not

in cell culture for an undetermined reason, possibly linked to biological availability. Unlike other inactive compounds, transient treatment with DL7 may exert a cell-specific effect that serves to diminish IAV titres. Thus, combined with the antagonistic behaviour of this compound in combination studies, we currently consider DL7 deprioritised for future development.

Irrespective of whether resistance may or may not arise to new M2-N31 specific compounds, the most clinically important observation from this study is that either lumenal and peripheral binding sites, or distinct M2 conformations, are viable drug targets that allow combinations of inhibitors to be used for therapy. Moreover, synergistic, rather than additive, antiviral effects were achieved for two of the four combinations tested. Interestingly, whilst it is clear how M2WJ332 and L1.1 binding at distinct sites might induce synergy, the combination of the former with another peripherally targeted compound, DP9, is conceptually more difficult. Notably, the pattern of synergy differed to the other combination as only occurring in the presence of lower M2WJ332 concentrations. We surmise that this may relate to the availability of four separate peripheral binding sites upon M2 complexes, as well as the greater potency of M2WJ332. L1.1 also displayed synergy when combined with Zanamivir. Hence, whilst the compounds herein represent the initial stages of hit identification for both binding sites, the indications are that further development will eventually enable double, triple, or even expanded therapeutic regimens upon inclusion of other agents. Such strategies are applied to antiviral treatment for other highly variable RNA viruses and there is growing consensus that such approaches represent the best way forward for influenza A virus. Furthermore, combinations should combat potential shortcomings for individual agents in terms of lower genetic barriers to potential resistance. Overall, future exploitation of both druggable sites/conformations within M2-N31 using specific inhibitors has considerable potential to rejuvenate this essential ion channel protein as a drug target, providing an important additional resource to combat emergence of future pandemics.

## Materials and Methods

### Peptide synthesis and reconstitution

Peptides (Eng195 M2-N31 CD and TM, M2-S31 CD) were purchased from Peptides International, and supplied lyophilised as a TFA salt. Both lyophilised and reconstituted peptides (1 mM in 100% methanol) were stored at -20˚C. For details, please see S1 Data.

H-RCSDSSDPLVIAA**S**IIGILHLILWITDRLFFKCIYRRFKYGLK-NH2 (S31 CD, 18-60aa)
H-RCSDSSDPLVIAA**N**IIGILHLILWITDRLFFKCIYRRFKYGLK-NH2 (N31 CD, 18-60aa)
H-RCSDSSDPLVIAA**N**IIGILHLILWITDRL-NH2 (N31 TM, 18-46aa)

### M2 activity and drug inhibition using liposome dye release assays

Unilammellar liposomes containing self-quenching concentrations of carboxyfluorescein were prepared as described[74]. M2 mediated CF release was assessed by incubating up to 50 nM peptide with 50 μM liposomes (determined by rhodamine absorbance). 100 μl assays were carried out in black-walled, flat-bottomed black-base 96 well plates (Grenier Bio One) with liposomes alone as a control for baseline fluorescence, liposomes + 5% v/v MeOH a negative solvent only control and 0.5% v/v Triton TX-100 (Triton) as a maximal positive control used for gain adjustment, setting a level of 90% fluorescence. CF release measured by increased fluorescence was taken as an indicator of peptide induced membrane permeability. A set of 30 or 63 readings (λex = 485 nm, λem = 520 nm) were made at 43 second (s) intervals using a FLUOstar Galaxy plate-reader (BMG Labtech). Assays were carried out at RT, with the plate kept on ice for 2–5 min after gain adjustment and while the peptide +/- compound were

added. All samples were repeated in duplicate on the plate and each experiment repeated three times (unless otherwise stated). Baseline readings of liposomes alone were subtracted from each of the wells when calculating end point results.

For compound inhibition assays, a maximum of 1% v/v DMSO was added to each reaction, with stock inhibitor concentrations diluted accordingly. Compounds or a DMSO control were pre-incubated with peptides for 20 min at RT prior to addition to chilled liposome suspensions. Baseline-subtracted end point values were averaged and normalised to the peptide + DMSO control. Values were then averaged across replicate experiments and standard errors calculated. Paired t-tests determined statistical significance between the peptide + DMSO control and compound treated reactions.

## Generation of Eng195 homology models based upon PDB 2RLF (octanol, bilayer)

Published M2 tetramer structures were downloaded from the website of the Research Collaboratory for Structural Bioinformatics (RCSB), protein data bank (PDB) http://www.rcsb.org/pdb/home/home.do). The solution NMR structure of A/Udorn/307/1972 H3N2 (PDB: 2RLF) [61] was used as a template for an *in silico* structural homology model of the E195 M2 conductance domain (A/England/195/2009). Prime and Maestro (Schrödinger) were used for homology modelling, with the model minimised in an octanol environment. Later, this homology model was re-minimised in a lipid membrane environment, using an Optimised Potentials for Liquid Simulations (OPLS) force field and used for docking of unbiased compound analogues. A second homology model using PDB 2L0J[85] as a template was made, using the SWISS-MODEL (ExPASy) web server and was minimised in an octanol environment using Maestro.

## *In silico* docking, compound selection and attrition

Two binding regions were defined, the lumen concentrated on pore lining residues at the N-terminal end of the TM domain (V27, A30, N31, I33 and (G34) and the periphery on lipid facing residues at the C-terminal of the TM domain (W41, I42, T43, D44, R45 and F48).

Open access small molecules libraries were used for an unbiased molecular binding study, with eHiTS (SimBioSys Inc.) used to dock compounds onto the two pre-defined binding regions of the Eng195 homology model. Ranked by eHiTS score the top 1000 hits, at each site, were manually assessed for their binding pose and drug-like qualities, resulting in seven predicted lumenal binding compounds L1-L7 and six peripheral binding compounds P1-P6 being selected for testing.

In addition, a biased screen was carried out utilising a rapid overlay of chemical structure (ROCS) approach and centred on compound D. ROCS (OpenEye Scientific) software was used and the top 1000 hits were docked against the homology model using eHiTS. Compound docking at both sites was validated using SPROUT (Keymodule Ltd.) software. A protocol of attrition was carried out to select DL and DP compounds, focussing on docking scores, molecular weight and specific interactions with the M2 tetramer. Full details are listed in (Section 3.5.1). Briefly, compounds were selected based on agreement between the two binding scores, molecular weight and specific interactions with the protein. Analogues of selected compounds were found via the online tool eMolecules (www.emolecules.com). Selected compounds were subsequently docked against the 2RLF-based E195 M2 homology model using eHiTS.

## Chemicals and molecular probes

M2WJ332 ((3S,5S,7S)-N-{[5-(thiophen-2-yl)-1,2-oxazol-3-yl]methyl}tricyle[3.3.1.1~3,7~] decan-1-aminium) was purchased from Avistron Chemistry Services. Rimantadine

hydrochloride ((RS)-1-(1-adamantyl)ethanamine) and zanamivir ((2R,3R,4S)-3-acetamido-4-(diaminomethylideneamino)-2-[(1R,2R)-1,2,3-trihydroxypropyl]-3,4-dihydro-2H-pyran-6-carboxylic acid) were purchased from Sigma, *NN*-DNJ from Toronto Biochemicals, all other compounds were purchased from ChemBridge. All commercially purchased lyophilised compound stocks were tested for purity using Liquid Chromatography Mass Spectrometry. Compounds were resuspended in dimethyl sulfoxide (DMSO) to 10–100 mM, aliquoted, and stored at –20 $^0$C, or –80 $^0$C for long term storage.

## Generation of gel-interface lipid bilayers

Electrical measurements of M2 peptides were performed using an interface bilayer formed between gel-gel interfaces, as shown in S3A Fig. Briefly, a homemade Teflon holder with two adjacent, connected wells (diameter 6 mm, and depth 8 mm) containing 150 μL agarose gel (1% w/v in 1 M KCl, 10 mM HEPES, pH 7.0) in one, and the same volume of buffer (1 M KCl, 10 mM HEPES, pH 7.0) in the other. 60 μL hexadecane & silicon oil (10:1) with 8 mg/ml DPhPC (1, 2-diphytanoyl-sn-glycero-3-phosphocholine, >99%, SIGMA) was next added into the holder on top of gel and buffer. A lipid monolayer formed at the oil/gel or oil/water interface within a few minutes. A drop of agarose gel (1% w/v in 1 M KCl, diameter ~ 1 mm) was attached onto a Ag/AgCl electrode (diameter 0.25 mm) and then immersed into the lipid-oil solution for at least 15 min. At the same time, another Ag/AgCl electrode was immersed into the KCl buffer. Following formation of a stable lipid monolayer on the surface of the gel droplet, this and the Ag/AgCl electrode were slowly lowered until making contact with the gel in the well, indicated by increased capacitance. Thus, a lipid bilayer was formed at the gel/gel interface (S3A Fig, insert). A period of several minutes was required to achieve stable capacitance prior to experimentation.

## Electrical measurements of interface lipid bilayers

After the formation of gel-gel interface bilayers, a range of voltages (up to several tens of mVs) was applied using an Axopatch 200 (Molecule Devices LLC, USA), and the current through the lipid bilayer was recorded. Resistance of the interface bilayer (calculated from the I-V curve) was generally 4 GΩ for an area of ~ 100 x 100 μm$^2$. This was checked by repeating the measurement three times by breaking-apart and re-forming the gel-gel interface lipid bilayer. Due to variation in the interface area, resistances were normalised to capacitance, i.e. equivalent to the unit area resistance. Normalised values were used to calculate mean conductance, enabling comparison between different experiments (Normalised Conductance = (ΔI/ΔV)/ Capacitance).

## Formation of M2 ion channels and inhibitor effects

Upon formation of stable interface lipid bilayers with resistance normalised to capacitance, M2 peptide in 100% MeOH was added to one of the wells containing KCl buffer. The maximum final concentration of methanol was limited to 5% (15 μL MeOH in 300 μL buffer & gel). Inhibitor assays were conducted using 200/400 (CD) - 1000 (TM) nM peptide, whereas single channel recordings utilised 1000 x lower concentration of peptide.

After adding peptides, > 1 hr resting time allowed full diffusion within the buffer and the gel. Note, the top electrode was placed close to the gel/buffer well boundary. Thus, peptides were required to diffuse for <1 mm rather than 6 mm, which reduced the time necessary for equilibration. Conductance of the lipid bilayer was then measured as described above, with significant increases caused by the addition of peptide. Inhibitors solubilised in DMSO were added to the buffer well following the formation of M2 ion channels with a maximum final

DMSO concentration of 1% v/v. S3 Fig describes typical experiments using M2 peptides and inhibitors.

## Plaque reduction assays for antiviral effects

Madin-Darby Canine Kidney (MDCK, obtained from the American Type Culture Collection (ATCC)) cells were infected with A/England/195/2009 (E195) influenza A virus. MDCKs were seeded 5 x$10^5$ / well of a 12 well plate 4 h prior to infection. IAV was diluted in SF media to a multiplicity of infection (MOI) of 0.01 or 0.001 and preincubated with compound diluted in DMSO for 30 min on ice, prior to a 1 h infection. Virus containing media was then removed and replaced with SF media with the addition of 1 µgml$^{-1}$ TPCK trypsin and compound before incubation of the producer plate at 37˚C, 5% $CO_2$. At 24 hpi the media was removed for titration. Virus containing supernatants were in SF media and dilutions of $10^{-1}$ to $10^{-4}$ were then used to infect fresh monolayers of MDCK for one hour. Media was then removed and a 3:7 mixture of 2% w/v agar (Oxoid Purified Agar) and overlay media (Modified eagles medium (MEM), 0.3% v/v BSA (fraction V), 2.8 nM L-Glutamine, 0.2% v/v NaHCO2, 14 mM HEPES, 0.001% v/v Dextran, 0.1x Penicillin and 0.1x Streptomycin) containing 2 µgml-1 TPCK trypsin was added to cells. Inverted plates were then returned to the incubator and incubated at 37˚C, 5% CO2 for 72 hr to allow plaques to form. After removal of the agar, cells were fixed in 2 ml 4% paraformaldehyde in PBS for 1 h, then stained with 1 ml of 1% v/v crystal violet solution for 5 min. Plaques were then counted and virus titres calculated. Each condition was carried out in triplicate within an experiment, with average titres and standard deviations calculated. Paired t-tests were carried out to assess statistical significance between DMSO control treated virus and other compounds.

To determine the half maximal inhibitory concentration ($IC_{50}$) of plaque formation, half-$log_{10}$ dilutions were made from 80 µM, down to 80 nM, except in the case of Zanamivir where $log_{10}$ dilutions were made from 80 µM, down to 80 pM.

## Chemical cross-linking of inhibitor treated M2 peptides

Crosslinking of Asn31 (N31) and Ser31(S31) M2 peptides was performed using Lomant's reagent (Dithiobis[succinimidyl proprionate (DSP, Pierce)). 5 µg of each M2 peptide was reconstituted in 300 mM DHPC/20 mM sodium phosphate buffer [pH 7.0] overnight in the absence or presence of 40 µM Rimantadine or *NN*-DNJ. Following reconstitution, DSP in DMSO was added to a final concentration of 75 µM. Reactions were incubated at room temperature for 15 min, prior to quenching with 200 mM Tris buffer pH 7.5. Cross-linked reactions were analysed on a 15% Tris-Glycine/6MUrea/SDS/PAGE gel.

## Cell viability and proliferation assays

MDCK cells were seeded at 2.5 x$10^4$ / well, into flat-bottomed clear 96 well plates (Corning Costar) and incubated overnight at 37˚C, 5% $CO_2$. Media was removed and cells were treated with dilutions of compounds up to 160 µM in cell culture media, with a final concentration of 0.2–0.8% v/v DMSO. Cells were incubated with the compound for 48–72 hr at 37˚C, 5% $CO_2$.

3-(4, 5-dimethyl-2-thiazolyl)-2, 5-diphenyl-2H-tetrazolium bromide (MTT), a tetrazolium dye, was used as a colourimetric indicator of cellular metabolism. A stock solution (2.5 mgml$^{-1}$) used to add 50 µg MTT to each of the wells and incubated at 37 $^0$C, 5% $CO_2$ for 4 h. Media was then replaced by 75 µl of DMSO and incubated at 37 $^0$C for 5 min prior to reading absorbance at 550 nm. Data was normalised to cells treated with the appropriate DMSO control (0.2–0.8% v/v). Cellular confluency after drug treatment was assessed using the IncuCyte

ZOOM (Essen Bioscience), with measurements of the percentage of well surface occupied by cells made under phase.

## Combination therapy plaque reduction assays and synergy calculations

Combinations of two M2 targeted compounds, or an M2 targeted compound and the licensed neuraminidase inhibitor Zanamivir were tested. Combination matrices were set up across a range from 0, 0.2, 1 and 5 μM, or based upon doses corresponding to multiples of the compound $IC_{50}$ value, with the 0, 0 combination as DMSO control. Each of the 16 conditions was run in triplicate within the experiment. Virus titres were normalised to the DMSO control, set to 100% relative virus titre.

The antiviral effect of combinations of inhibitors were evaluated using two methods: 1) MacSynergy utilising the Bliss independence model, where values <25 at 95% confidence indicate synergy that is insignificant, 25–50 minor but significant, 50–100 moderate and >100 strong synergy. 2) CompuSyn (ComboSyn, Inc., www.combosyn.com) [94] software, which uses the Loewe additivity model where combination indexes (CI) indicate synergy (<0.9), antagonism (>1.1) and additive (0.9–1.1) interactions. Data presented in Fig 7 are from MacSynergy, Compusyn data was comparable and is available upon request.

## Selection in culture using M2-specific compounds and plaque purification of single variants

For serial passage using increasing compound concentrations, MDCK cells were seeded into 6 well plates 4 h before the initial infection with Influenza virus was carried out as described above at an MOI of 0.001, with 2.5 μM compound. At 24 hpi virus containing media was removed, 1/10th volume was used to infect freshly seeded MDCK cells, as a blind passage and the remainder was snap frozen. This process was repeated, each time increasing the concentration of compound present in the media two-fold, until 80 μM was reached. At selected time points the titre of viral supernatants was determined via plaque assay. These supernatants could then be used at MOI 0.001, with fresh 80 μM compound, in subsequent infections.

For plaque purification, MDCK cells were seeded in 12 well plates 4 h before infection. Virus was diluted 1:250 in SF media containing between 5 and 80 μM compound and used to infect cells for 1 h, before cells were set under overlay media, containing 2 μg/ml TPCK, 0–80 μM compound and agar. At 72 hpi, agar plugs were picked and placed in 300 μl SF media for 2 h prior to it being used to infect fresh MCDK cells, in the presence or absence of compound (5–80 μM) for 1 h at 37 $^{0}$C, 5% $CO_2$. Once infectious supernatant was removed, it was replaced with SF media + 1 μg/ml TPCK and 0–80 μM compound and plates returned to 37 $^{0}$C, 5% $CO_2$. Once > 40% CPE was observed, infectious supernatant was clarified, prior to vRNA extraction.

## Extraction, purification and sequencing of virion RNA (vRNA)

vRNA was extracted from clarified supernatants using a QIAamp Viral RNA Mini Kit (QIAGEN) according to the manufacturer's instructions. Resultant eluted vRNA was kept at –20 $^{0}$C for short term storage, or transferred to -80 $^{0}$C for long term storage. vRNA was synthesised into first strand cDNA using SuperScript III (SSCIII) (Invitrogen) and a Eng195 segment 7 specific forward primer (sequences available upon request). A negative control of vRNA but no SSCIII was included in each experiment.

cDNA was amplified via polymerase chain reaction (PCR) using the proof reading Phusion high fidelity (HF) polymerase (Phusion) (New England Biolabs). Reactions were heated to 98 $^{0}$C for 30 s, followed by 35 cycles of the following steps; denaturation at 98 $^{0}$C for 10 s,

annealing at 48 $^{0}$C for 30 s and extension at 72 $^{0}$C for 40 s and a final incubation at 72 $^{0}$C for 7 min. Amplified cDNA was purified using a QIAquick PCR Purification Kit (QIAGEN) according to the manufacturer's instructions, with eluted DNA concentrations determined using a nanodrop spectrophotometer and DNA visualised by Tris-acetate-EDTA buffered agarose gel electrophoresis. Samples were stored at –20 $^{0}$C.

### Direct sequencing of virus-derived cDNA

Standard dsDNA sequencing was conducted using the Mix2Seq kit (Eurofins Genomics), with forward internal primer Eng195_s7_Fint (5'-GGCTAGCACTACGGC-3') or reverse primer Flu_s7_R2 (5'-AGTAGAAACAAGGTAGTTTTTTACTCTAGC-3').

### Next Generation sequencing of total genomic viral RNA

vRNA was reverse transcribed by Superscript III (Invitrogen) and amplified by Platinum Taq HiFi Polymerase (Thermo Fisher) and influenza specific primers[95] in a 1-step reaction. Library preparation was performed using a Nextera kit (Illumina). Libraries were sequenced on an Illumina MiSeq using a v2 kit (300-cycles; Illumina) giving 150-bp paired end reads. Reads were mapped with BWA v0.7.5 and converted to BAM files using SAMTools (1.1.2). Variants were called using QuasiBAM, an in-house script at Public Health England.

### Supporting information

**S1 Fig. Characterisation of liposome carboxyfluorescein (CF) release assay for M2 peptides. A.** titration of biological activity for Eng195 M2 CD (N31/S31) and TM peptides. TM peptides form channels less efficiently compared with CD, thereby requiring 320 nM compared with 40 nM for equivalent biological activity at each experimental condition. **B.** Biological activity of M2 peptides in liposome release assay is responsive to acidic pH. End-point assays were conducted in alternate pH buffers prior to removal of liposomes by ultracentrifugation and re-buffering of reactions to restore CF fluorescence, as described previously. **C.** Disruption of M2 oligomerisation by alkylated imino-sugars. M2-N31 and M2-S31 CD peptides were incubated with liposomes in the presence of 40 μM rimantadine or $N$N-DNJ, prior to the addition of DSP (Lomant's reagent). Resultant M2 complexes were visualised using SDS-Urea gel electrophoresis.
(TIF)

**S2 Fig. Activity of prototypic compounds showing specific M2-N31 activity versus Eng195 in vitro in MDCK plaque reduction assays.** Compounds present at 80 μM both during infection (MOI of 0.01 pfu/cell) and through a 24 hr incubation in producer cells prior to titration of secreted infectivity. Rimantadine (80 μM) negative control was included. Data are representative of three technical repeats ($^{*}$ p≤0.05, paired student t-test).
(TIF)

**S3 Fig. Interface lipid bilayer measurements of M2 peptide activity in the presence or absence of inhibitors. A.** Organisation of the in-house Teflon holder rig used to create gel-gel interface bilayers as described in materials and methods. Inset shows orientation and formation of lipid bilayers. **B.** Example characterisation of bilayers harbouring M2 peptides showing the of current values at different voltages (left panel), plotted as a histogram fitted by multiple Gaussian peaks to determine the mean current values at different voltages (middle), allowing an I-V curve (right) to be plotted for use in calculating normalised bilayer conductance. **C.** Single channel recordings of M2-N31 CD peptides at voltages of +/- 50 mV. **D.** Normalised conductance of M2-N31 CD versus TM peptides (2000 nM) taken from three independent

experiments. Error bars represent standard error of the mean with $p$ values determined using the student T-test (* $p \leq 0.05$, ** $p \leq 0.01$). **E.** Inhibition of TM peptides (2000 nM) using L1.1 (1333 nM) taken from three independent experiments. Error bars represent standard error of the mean with $p$ values determined using the student T-test (* $p \leq 0.05$, ** $p \leq 0.01$).
(TIF)

**S4 Fig. Predicted interactions between novel compounds and Eng195 homology model. A.** Predicted interactions between M2WJ332 and the Eng195 (E195) homology model when docked into the lumen (left) and the periphery (right). Docking studies conducted using Glide. **B.** As for A, but detailing L1.1 (Lumen), DL7 (lumen) and DP9 (periphery).
(TIF)

**S5 Fig. Metabolic activity of M2-N31 inhibitor treated cells.** Compounds showing significant activity versus M2-N31 *in vitro* were assessed for potential cytotoxic effects. MDCK cells were incubated with increasing compound concentrations (20–160 μM) for 24 h prior to assessing metabolic activity via MTT assay. Results are representative of at least two biological repeats, each containing four technical repeats. Error bars are standard deviations within one biological repeat. Compounds showing any sign of impaired metabolic effects at 40 μM or greater were discounted from virological screens.
(TIF)

**S6 Fig. Plaque purification and expansion of Eng195 bulk cultures from SP5.** Normalised supernatants were diluted and plated onto naïve cells (triplicates in a 12-well plate) in the presence of increasing inhibitor concentrations for 1 h, after which media was replaced by media/soft agar containing the same inhibitor concentrations. Resultant plaques (mean from triplicate wells) were counted, plotted as a function of inhibitor concentration, and fitted to exponential curves in Excel with $R^2$ values as shown.
(TIF)

**S7 Fig. Determination of Zanamivir IC$_{50}$ versus Eng195 in MDCK culture.**
(TIF)

**S8 Fig. Predicted interactions of L1.1, DP9, DL7 and M2WJ332 with lumenal and peripheral binding sites in 2L0J-based Eng195 homology model.**
(TIF)

**S1 Data. Information regarding commercial peptide synthesis provided by Peptides International, including purity assessment.**
(PDF)

**S1 Table. Predicted interactions of adamantane compounds (rimantadine, compound D) with 2RLF-based Eng195 M2 homology models.**
(DOCX)

**S2 Table. Properties of M2-N31 inhibitors in vitro and in Eng195 virus culture.**
Table summarising the compounds arising from *in silico* HTS targeting peripheral and/or lumenal binding sites. Compounds were screened for activity in liposome dye release at least twice, with several progressed into toxicity and cell culture studies. Compound names and Chembridge IDs are shown, along with molecular structures and predicted binding (L: Lumen; P: Periphery) based upon E-Hits/Sprout programmes. M2 activity in the presence of compounds (40 μM) is shown for M2-N31/S31 peptides corresponding to the CD or TM regions of the protein. Observed site preferences (Obs L/P) are based upon relative inhibition of M2 CD/TM activity, with "?" indicating potential binding to lumen or partial peripheral

binding site based upon compound titrations; bold text indicates differences from predicted binding. Several compounds were tested versus Eng195 in culture (80 μM) and the order of magnitude titre reduction across at least three assays is shown. Finally, IC$_{50}$ was determined for four compounds selected for synergy experiments.
(DOCX)

**S3 Table. Complete list of mutations observed during NGS analysis of serial drug selection cultures (passage 5 and 14).**
(XLSX)

## Author Contributions

**Conceptualization:** Wendy S. Barclay, Richard Foster, Stephen Griffin.

**Data curation:** Daniel H. Goldhill, Wendy S. Barclay, Richard Foster, Stephen Griffin.

**Formal analysis:** Daniel H. Goldhill, Katie Simmons, Jason R. Schnell, Richard Foster, Stephen Griffin.

**Funding acquisition:** Paul Targett-Adams, Jason R. Schnell, Richard Foster, Stephen Griffin.

**Investigation:** Claire Scott, Jayakanth Kankanala, Toshana L. Foster, Daniel H. Goldhill, Peng Bao, Katie Simmons, Marieke Pingen, Matthew Bentham, Elizabeth Atkins, Eleni Loundras, Jolyon K. Claridge, Joseph Thompson, Peter R. Stilwell, Ranjitha Tathineni, Stephen Evans, Richard Foster, Stephen Griffin.

**Methodology:** Jayakanth Kankanala, Toshana L. Foster, Peng Bao, Katie Simmons, Elizabeth Atkins, Ruth Elderfield, Joseph Thompson, Peter R. Stilwell, Stephen Evans, Wendy S. Barclay, Richard Foster, Stephen Griffin.

**Project administration:** Stephen Griffin.

**Resources:** Clive S. McKimmie, Paul Targett-Adams, Stephen Evans, Wendy S. Barclay, Richard Foster, Stephen Griffin.

**Software:** Richard Foster.

**Supervision:** Ruth Elderfield, Clive S. McKimmie, Jason R. Schnell, Graham P. Cook, Stephen Evans, Wendy S. Barclay, Richard Foster, Stephen Griffin.

**Validation:** Richard Foster, Stephen Griffin.

**Visualization:** Richard Foster, Stephen Griffin.

**Writing – original draft:** Stephen Griffin.

**Writing – review & editing:** Jason R. Schnell, Graham P. Cook, Wendy S. Barclay, Richard Foster, Stephen Griffin.

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
