## [Decision Letter · Decision Letter 0]

14 Nov 2019

Dear Dr Griffin,

Thank you very much for submitting your manuscript "Site-directed M2 proton channel inhibitors enable synergistic combination therapy for rimantadine-resistant pandemic influenza" (PPATHOGENS-D-19-01729) for review by PLOS Pathogens. Your manuscript was fully evaluated at the editorial level and by independent peer reviewers. The reviewers appreciated the attention to an important problem, but raised some substantial concerns about the manuscript as it currently stands. These issues must be addressed before we would be willing to consider a revised version of your study. We cannot, of course, promise publication at that time.

We therefore ask you to modify the manuscript according to the review recommendations before we can consider your manuscript for acceptance. Please carefully consider each of the critiques and address each point raised. In light of the comments of Reviewer 1, we ask that you pay particular attention to ensuring that the data rigorously support the conclusions drawn, and include additional experiments needed to achieve this goal.

(1) A letter containing a detailed list of your responses to the review comments and a description of the changes you have made in the manuscript. Please note while forming your response, if your article is accepted, you may have the opportunity to make the peer review history publicly available. The record will include editor decision letters (with reviews) and your responses to reviewer comments. If eligible, we will contact you to opt in or out.

(2) Two versions of the manuscript: one with either highlights or tracked changes denoting where the text has been changed; the other a clean version (uploaded as the manuscript file).

Additionally, to enhance the reproducibility of your results, PLOS recommends that you deposit your laboratory protocols in protocols.io, where a protocol can be assigned its own identifier (DOI) such that it can be cited independently in the future. For instructions see http://journals.plos.org/plospathogens/s/submission-guidelines#loc-materials-and-methods

We hope to receive your revised manuscript within 60 days. If you anticipate any delay in its return, we ask that you let us know the expected resubmission date by replying to this email. Revised manuscripts received beyond 60 days may require evaluation and peer review similar to that applied to newly submitted manuscripts.

[LINK]

Sincerely,

Anice C. Lowen

Associate Editor

PLOS Pathogens

Andrew Pekosz

Section Editor

PLOS Pathogens

Kasturi Haldar

Editor-in-Chief

PLOS Pathogens

orcid.org/0000-0001-5065-158X

Grant McFadden

Editor-in-Chief

PLOS Pathogens

orcid.org/0000-0002-2556-3526

Reviewer's Responses to Questions

**Part I - Summary**

Reviewer #1: Strength:

1) it raised a scientific controversy which will attract significant attention from the scientific community of M2.

2) The resistance selection experiment is well executed and the results make sense.

Weakness:

1) The conclusion was not supported by stringent experimental data.

2) It is important to include at least two independent assays to validate the proposed mechanism of action. In this study, the only experiment conducted was the liposome flux assay.

Reviewer #2: More than 95% of the currently circulating influenza A viruses carry the amantadine and rimantadine-resistant M2-S31N proton channel. For this reason the topic of the current manuscript, the study of the M2-S31N channel and the design of M2-S31N inhibitors taking into account the controversy surrounding the two potential drug binding sites for Amt, is of the greatest interest.

The overall quality of the manuscript is excellent. The paper is concise, very well written and the experimental design is straightforward. The introduction is clear and up-to-date. Both the computational and the biology parts of the work have been skillfully conducted and the results are clearly presented and discussed.

Reviewer #3: Scott et al report extensive work leading to the observation of a novel anti-influenza compound, L1.1, two additional compounds of possible value, DL7 and DP9, and the discovery of resistance formation to a previous invention M2WJ332 via M2 V27A.

The means of identifying the novel compounds are dubious: in silico screening of two established M2 binding sites, the accepted luminal amantadine binding site and a peripheral binding site in the micellar M2 CD structure that is largely abrogated in the membrane protein structure, and carboxyfluorescein release from M2-proteoliposomes, which has not been calibrated against electrophysiology and depends on the assumption that a bulky dye is somehow transported by the extremely narrow M2 channel.

Nevertheless, the dye release assay has been used extensively by the authors for assessment of the broader (OuYang et al, 2013, Nature 498:521-525) P7 channels in hepatitis C, gave internally consistent results (M2WJ332 blocked the N31 M2 transport and not the S31 M2 transport, as found in electrophysiology), and most importantly led to the discovery of compounds with outstanding potency against influenza infections in cell culture.

Furthermore, the proteoliposome approach allowed easy comparison of dye release properties of M2 TMD and M2 CD, which is novel. However, the authors may have been too accepting of the assumption that compounds that reduce dye release in the CD but not the TMD must be blocking peripherally, as this assumption has not been independently validated.

The study and article are very well-crafted and references the current literature accurately and thoroughly. The scientific methodology is similarly state-of-art and thorough.

The only general concern is that perhaps the results are slightly over-interpreted. Carboxyfluorescein transport may be mediated by M2 monomer insertions, given that the peptide is added to the dye-containing vesicles post hoc. The designations of peripheral and luminal binding based on dye release may be based on false assumptions. The DL7-resistant virus should have shown mutations too. L1.1 seems to be too large to get past the V27 barrier and get into the luminal site in M2. The lack of resistance developing in the presence of L1.1 could mean that it’s blocking a physiologically critical site in M2; or, just that L1.1 is blocking some other key viral or host process, and that the dye release result is a red herring. Likewise for the lack of M2 sequence change in 9/9 sequences of plaque-purified L1.1-passaged virus.

**Part II – Major Issues: Key Experiments Required for Acceptance**

Reviewer #1: This study by Griffin et al. is interesting. It brought a new scientific controversy regarding the drug binding site(s) of the influenza A virus M2 proton channel. In this study, the authors used liposome flux assays to screen non-adamantane based inhibitors for the M2-S31N mutant, which is the predominant drug resistant mutant in circulating influenza A viruses. In this assay, they incorporated two M2 constructs, the transmembrane domain (TM) which only contains the luminal drug-binding site, and the conductance domain (CD) which contains both the luminal drug-binding site and the peripheral drug- binding site. They first attempted to prove the validity/feasibility of the liposome flux assay using known M2 channel blocker rimantadine plus a few other compounds that may or may not be relevant. Super singly, they claimed that a previously well-characterized compound WJ332 actually inhibited M2 through targeting the peripheral site, not the luminal site. Nevertheless, the authors then conducted a virtual screening to identify novel compounds that can bind to either the luminal site or the peripheral site or both. Four non-adamantane compounds were identified and further confirmed in the liposome flux assay. The antiviral activity of two compounds L1.1 and DL7 were further confirmed in viral replication assay. Next, the authors performed resistance selection experiments and found no M2 mutations for L1.1, but V27A mutant for WJ332. Finally, compound L1.1 was shown to have synergistic antiviral activity with either WJ332 or zanamivir.

Although the authors claimed interesting and somewhat controversy findings, the conclusions are not supported by the experiment results. Many of the experiments are poorly designed and lacks vigorous controls.

Major comments are:

1) The accuracy and reliability of the liposome flux assay is in doubt. There is no stringent calibration/validation for this assay. For example:

a. In Fig. 1B, rimantadine only inhibited 50% of channel activity of M2-WT (31) at concentrations up to 40 uM. This is in direct contrast to the efficacy of rimantadine. Rimantadine inhibits M2-WT containing viruses with submicromolar efficacy and blocks M2-WT channel with single micromolar efficacy in electrophysiological assays.

b. In Fig. 1B, compounds D, H, J were shown to potently inhibit both M2-WT and M2-S31N in the liposome flux assay; therefore, these three compounds should inhibit both M2-WT and M2-S31N containing viruses. However, no such data was provided. Similarly, antiviral assays should be performed for compounds B and K to test whether they only inhibit M2-WT containing viruses, but not M2-S31N containing viruses.

c. In Fig. 3C, WJ332 only inhibited ~40% of channel activity of the N31 CD peptide in the liposome flux assay when tested at up to 40 uM. This is in direct contrast to the efficacy of WJ332 in electrophysiological assay and antiviral plaque assay. WJ332 had an IC50 of 16 uM in two-electrode voltage clamp assay and an EC50 of 153 nM in antiviral plaque assay (PNAS January 22, 2013 110 (4) 1315-1320).

d. Both the N31 CD and N31 TM peptides contain the luminal drug-binding site, which means compounds that bind to the luminal site should show equal or more potent channel inhibition for the N31 CD construct than the N31 TM construct (for example, compounds that happen to bind to both luminal and peripheral sites). However, this is not the case for compound P6.1 (Fig. 4B), there must be something seriously wrong with this assay.

2) The claim that WJ332 is a peripheral M2 inhibitor is premature. The ONLY piece of experimental data the author provided to support this claim is in Fig. 3C, in which WJ332 appeared to only slightly inhibit N31 CD but not N31 TM. To confirm/validate a proposed drug binding mechanism, here are the general approaches: 1) Determine the drug-protein bound structure. The structure of WJ332 in complex of M2-S31N was solved by solution NMR (PDB: 2LY0), which showed luminal drug binding. In addition, the drug binding of similar compounds was determined by either solution NMR or solid state NMR (J Am Chem Soc 135, 9885-9897; J. Am. Chem. Soc. 136, 17987-13995). 2) Mutagenesis. Mutations on the residues located at the proposed drug-binding site should confer drug resistance in both electrophysiological assay as well as cellular antiviral assay. Mutations (V27A, N31S, N31D, I32T) located at the luminal drug-binding site have been shown to confer resistance against M2WJ332 and similar compounds (Mol. Pharmacol. 2016, mol.116.105346; Antiviral Res. 2018, 153, 10-22.). In this study, the authors also selected V27A mutant with the drug selection pressure of WJ332, which supports the luminal binding mode. 3) Structure-activity relationship studies. Numerous analogs of WJ332 have been synthesized and tested, and the SAR is consistent with the luminal drug-binding mode.

Since a NMR expert, Jason Schnell, was listed as a coauthor, why don’t they solve the solution NMR structure of M2-S31N CD construct in complex with L1.1, DL7, and DP9 if they truly believe these molecules are M2 inhibitors? Maybe they should also repeat the NMR experiment with WJ332 and prove that WJ332 binds to the peripheral site but not the luminal site! It appears the scientific controversy never settles, the first one is related to amantadine binding to M2 WT, now it comes to WJ332/L1.1 binding to M2 S31N. This is exciting!

3) Following the 2nd point, the author failed to provide any relevant experimental results, other than the questionable liposome flux assay, to support the proposed luminal binding mode or the peripheral binding mode. Will mutations located at either the luminal or the peripheral binding sites of L1.1 confer resistance against this compound in both liposome flux assay and antiviral assay?

4) No M2 mutant was found during the serial viral passage experiment with L1.1, which means this compound inhibit influenza A virus through a different mechanism of action other than targeting M2 directly!

5) M2 is a proton channel and almost all of the potent M2 channel blockers contain a positively charged amino group or equivalent, which serves as a mimetic of the conducting hydronium ion. However, all three hits identified in this study L1.1, DL7, and DP9 are neutral compounds. The proposed binding modes shown in Fig. S2 do not seem to make sense. The interactions appeared to be mainly driven by hydrophobic interactions.

6) Compound L1.1 should also be tested against other influenza A and B viruses to make sure it does not have non-specific antiviral activity. WJ332 and related compounds have been extensively tested against multiple influenza A and B viruses.

7) In the combination therapy experiment, how could two compound that compete for the same drug-binding site have synergistic effect? This makes no sense.

8) How could L1.1 be classified as a luminal binding compound? There is no difference in N31 CD and N31 TM inhibition as shown in Fig. 4B.

Reviewer #2: (No Response)

Reviewer #3: (No Response)

**Part III – Minor Issues: Editorial and Data Presentation Modifications**

Reviewer #1: Minor comments:

1) Introduction: M2 is a 97-residue long peptide, not 96-residue.

2) Introduction: it states “ensuing controversy remains”. This is an inappropriate description. Since the two Nature papers published in 2008, numerous studies have been conducted to validate the pharmacology relevant drug binding sites of M2, including Dr. Chou’s own study. I think it is fair to claim that the controversy has settled.

3) Why are compounds L, B, and K called amiloride-related compounds? There is absolutely no structural similarity.

4) There is no characterization for the M2 peptides.

5) Molecule in Fig. 3D is wrong. Anyone with chemistry background should realize that the molecule shown in Fig. 3D is not the same molecule in Fig. 3B. How could isoxazole be protonated?

Reviewer #2: (No Response)

Reviewer #3: Table 1 could be consigned to SI, except for the first four rows, as long as the selection process is better explained in the Table legend.

Figure 6b: What about the others? No change in Rim because, even at 80 uM it is ineffective. No effect of Zan on M2. DMSO (what %?) passaging should not affect M2 very quickly, right? Didn't DL7 produce resistance yet? It has resistance by P14 according to fig 6D..., but apparently it hasn't developed yet at P6 according to fig 6C. What about the DL7 plaques?

Figure 6c: I don't understand the second sentence of the 6c legend, particularly the second half: "Quantification of SP7 output titres generated from infection by the subsequent passage with quantified SP6 innoculae." Is it supposed to mean: "Quantification of SP7 output titres after infection by equivalent doses of SP6 innoculae?"

Results describing Figure 6c: What is meant by the evolutionary bottleneck applied during passaging?

Results describing deep sequencing: What was the range of conditions examined? Complete genome? If so, were there any other changes? All 5 compounds used for passaging? It’s odd that there were no changes for cultures with DL7, which showed resistance formation in figure 6D. What is the zanamivir mutation in NA?

Results describing synergy: It would be appropriate to acknowledge the “multiple comparisons” problem. Considering all 5 compounds tested, there are 24 possible synergy combinations you could have tested for. What is the likelihood that one would see two appearing to show synergy at the level observed and two showing antagonism out of that many tests even if there is no synergy or antagonism, i.e. randomly? It may be very high.

Discussion starting at the bottom of page 20 with “Eng195 chronic culture…”: This paragraph works hard to rationalize the peripheral binding site for WJ332, but given the beliefs of the field, the 2LY0 structure, and the resistance to S31N/V27A, it might be good to acknowledge that the failure to block the N31 TM is not necessarily pathognomonic for binding location.

PLOS authors have the option to publish the peer review history of their article (what does this mean?). If published, this will include your full peer review and any attached files.

Reviewer #1: No

Reviewer #2: No

Reviewer #3: Yes: Prof. David D. Busath, M.D.

---

## [Decision Letter · Decision Letter 1]

3 Jun 2020

Dear Dr Griffin,

Thank you very much for submitting your manuscript "Site-directed M2 proton channel inhibitors enable synergistic combination therapy for rimantadine-resistant pandemic influenza" for consideration at PLOS Pathogens. As with all papers reviewed by the journal, your manuscript was reviewed by members of the editorial board and by independent reviewers. In this case, the revised manuscript has been reviewed by two of the prior reviewers. Based on the reviews, we are likely to accept this manuscript for publication, providing that you modify the manuscript according to the review recommendations.

While the opinions of the two reviewers differ, each raised valid concerns regarding the revised text. We ask that you modify the text to address the remaining concerns of Reviewer 1, including how the present data relate to prior characterization of the binding site of M2WJ332 by NMR; role of position 27 in binding and resistance; interpretation of the electrophysiology data in Fig S3; implications of failure to select resistance to DL7; and rationale for the synergy noted when M2WJ332 is combined with either L1.1 or DP9. As suggested by Reviewer 3, we also ask that you add commentary regarding the relationship of the present findings to those of Chizhmakov et al. 1996, and encourage you to consider removing Figures S3C and S3E.

Sincerely,

Anice C. Lowen

Associate Editor

PLOS Pathogens

Andrew Pekosz

Section Editor

PLOS Pathogens

Kasturi Haldar

Editor-in-Chief

PLOS Pathogens

orcid.org/0000-0001-5065-158X

Michael Malim

Editor-in-Chief

PLOS Pathogens

orcid.org/0000-0002-7699-2064

Thank you for your revisions. The manuscript has been reviewed by two of the prior reviewers. While their opinions differ, these reviewers each raised valid concerns regarding the revised text. We ask that you modify the text to address the remaining concerns of Reviewer 1, including how the present data relate to prior characterization of the binding site of M2WJ332 by NMR; role of position 27 in binding and resistance; interpretation of the electrophysiology data in Fig S3; implications of failure to select resistance to DL7; and rationale for the synergy noted when M2WJ332 is combined with either L1.1 or DP9. As suggested by Reviewer 3, we also ask that you add commentary regarding the relationship of the present findings to those of Chizhmakov et al. 1996, and encourage you to consider removing Figures S3C and S3E.

Reviewer Comments (if any, and for reference):

Reviewer's Responses to Questions

**Part I - Summary**

Reviewer #1: Please check the comments in the attachment.

Reviewer #3: Griffin and colleagues have responded cogently to virtually all of the reviewers' comments, and have improved the paper appropriately. The biggest addition is that of bilayer experiments with the peptides.

They are encouraging in one way and discouraging in two ways.

On one hand, they appear to illustrate drug blockability of peptide-induced conductance for M2WJ332 vs M2-N31, esp CD.

On the other hand, they must not have been successful for the novel drugs (not mentioned).

Furthermore, the supplemental figure illustrates a decades-old dilemma with M2 expression in planar bilayers: namely that although the channels were shown to be exquisitely selective and to give sub-fA currents for protons in mouse erythroleukemia cells (Chizhmakov et al 1996), and are readily blocked in transfected Xenopus oocytes (e.g. AMT blocks WT), they have not been found to be blocked (again, WT by AMT), had pA currents, and in the reviewer's lab (unpublished) to be non-selective among cations. This discrepancy has traditionally been explained (by the reviewer at least) as a possible result of modest (unmeasurable) surface tension in suspended bilayers deriving from the suspension from the torus, but the new results of single channel currents in the supported bilayers presented in Figure S3C rule out the explanation of modest surface tension strain induced channel over-opening because supported bilayers would be expected to have ~zero surface tension, much like vesicles and cells.

The blockability illustrated in the bar plots of Figure 3E and 3F are very gratifying, and they claim to represent 3 experiments with small standard deviations. The level of conductance decline displayed in Figure S3E is gratifying. One wonders if over the course of the block process the single-channel level was attained and then complete block ensued. Figure S3E shows only a one-second trace, so the reader can't tell if the "blocked" trace was taken between single channel currents, or if all channels were blocked. In any event, the pA currents in Figure S3C recall haunting years of controversy in this reviewer's lab.

I think the authors would be wise to remove Figures S3C and S3E - they arouse unfriendly reactions unnecessarily. Also, the x-axis in Figure S3B needs labeling and the and peaks should be labeled with associated membrane potentials, instead of just numbered. (Note that peak number 3 falls off the left of the plot into the axis label where there is no peak). If the authors want to state that they observed pA channels that they can't explain given the high selectivity and low maximum single channel conductance observed by Chizhmakov et al 1996, that would maintain scientific integrity AND make the case for channel structure dynamics that they want to suppose underlies dye permeation. I note, however, that such channel dynamics and corresponding fluctuations in selectivity would argue against the Chizhmakov findings, which should also be duly noted.

Finally, it is troublesome that the authors suppose that the gel compartment equilibrates in peptide and drug concentration on the one-hour time scale. With typical diffusion coefficients in unstirred gel, Fick's laws would predict weeks, not hours, for the equilibration time of a 6 mM diameter well loaded from the side.

**Part II – Major Issues: Key Experiments Required for Acceptance**

Reviewer #1: Please check the comments in the attachment.

Reviewer #3: (No Response)

**Part III – Minor Issues: Editorial and Data Presentation Modifications**

Reviewer #1: Please check the comments in the attachment.

Reviewer #3: Legend S2 needs number correction (it is called S3).

Significant figures for numbers need to be limited appropriately in the new material, and probabilities given as hypothesis test values (e.g. <0.05, as if the significance level threshold, alpha, had been preset).

PLOS authors have the option to publish the peer review history of their article (what does this mean?). If published, this will include your full peer review and any attached files.

Reviewer #1: No

Reviewer #3: Yes: Prof. David D. Busath, M.D.
---

## [Editor Report · Decision Letter 2]

19 Jun 2020

Dear Dr Griffin,

We are pleased to inform you that your manuscript 'Site-directed M2 proton channel inhibitors enable synergistic combination therapy for rimantadine-resistant pandemic influenza' has been provisionally accepted for publication in PLOS Pathogens.

Best regards,

Anice C. Lowen

Associate Editor

PLOS Pathogens

Andrew Pekosz

Section Editor

PLOS Pathogens

Kasturi Haldar

Editor-in-Chief

PLOS Pathogens

orcid.org/0000-0001-5065-158X

Michael Malim

Editor-in-Chief

PLOS Pathogens

orcid.org/0000-0002-7699-2064
---

## [Editor Report · Acceptance letter]

15 Jul 2020

Dear Dr Griffin,

We are delighted to inform you that your manuscript, "Site-directed M2 proton channel inhibitors enable synergistic combination therapy for rimantadine-resistant pandemic influenza," has been formally accepted for publication in PLOS Pathogens.

Best regards,

Kasturi Haldar

Editor-in-Chief

PLOS Pathogens

orcid.org/0000-0001-5065-158X

Michael Malim

Editor-in-Chief

PLOS Pathogens

orcid.org/0000-0002-7699-2064